# Condensation and asymmetric amplification of chirality in achiral molecules adsorbed on an achiral surface

Huiru Liu [1,2,5], Heping Li [3,5], Yu He[1,2], Peng Cheng [1,2], Yi-Qi Zhang[1,2], Baojie Feng [1,2], Hui Li [3] ✉, Kehui Wu [1,2,4] ✉ & Lan Chen [1,2,4] ✉

The origin of homochirality in nature is an important but open question. Here, we demonstrate a simple organizational chiral system constructed by achiral carbon monoxide (CO) molecules adsorbed on an achiral Au(111) substrate. Combining scanning tunneling microscope (STM) measurements with density-functional-theory (DFT) calculations, two dissymmetric cluster phases consisting of chiral CO heptamers are revealed. By applied high bias voltage, the stable racemic cluster phase can be transformed into a metastable uniform phase consisting of CO monomers. Further, during the recondensation of a cluster phase after lowering down bias voltage, an enantiomeric excess and its chiral amplification occur, resulting in a homochirality. Such asymmetry amplification is found to be both kinetically feasible and thermodynamically favorable. Our observations provide insight into the physicochemical origin of homochirality through surface adsorption and suggest a general phenomenon that can influence enantioselective chemical processes such as chiral separations and heterogeneous asymmetric catalysis.

Chirality is a fundamental concept in physical and life sciences. In chemistry, the enantiomers of chiral molecules often exhibit significantly different or even opposite physical properties, suggesting the possibility of valuable applications such as in optical and electronic devices[1–7]. In general, due to energy degeneracy, it is quite difficult to use non-chiral physicochemical methods to separate chiral enantiomers. However, homochirality is ubiquitous in nature, and is manifested in many biomolecules, such as amino acids, sugars, proteins, and DNA. The two enantiomers of chiral drugs sometimes show remarkable differences in their biological effects. The basic principles behind chiral condensation and separation have been debated[8,9] for decades and are still controversial. The source of homochirality is still an open question.

It is widely accepted that the origin of homochirality in the prebiotic environment is the spontaneous formation of a small enantiomeric excess followed by processes that amplify the excess[10]. Numerous previous studies have reported that chiral amplification of chiral molecules on achiral substrates can be due to self-organization[11], self-disproportionation[12], autocatalysis[13,14], enantioselective crystallization[15–17], or polymerization[18–20]. Furthermore, the chirality of achiral molecules induced by chiral surfaces[21–27] has also been reported and supported by many theoretical works:[28–30] the chiral chemical potential from a substrate can induce symmetry breaking and enhance chiral condensation after molecular adsorption. Yun et al. also demonstrated that the adsorption-induced auto-amplification of enantiomeric excess can occur on achiral surfaces due to the formation of homochiral adsorbate clusters, a phenomenon known as the "sergeants and soldiers" effect[12]. Most of the above-mentioned research is based on the hypothesis of a chiral conservation principle. It is still unknown whether any other

[1]Institute of Physics, Chinese Academy of Sciences, Beijing 100190, PR China. [2]School of physics, University of Chinese Academy of Sciences, Beijing 100049, PR China. [3]Beijing Advanced Innovation Center for Soft Matter Science and Engineering, Beijing University of Chemical Technology, Beijing 100029, PR China. [4]Songshan Lake Materials Laboratory, Dongguan, Guangdong 523808, PR China. [5]These authors contributed equally: Huiru Liu, Heping Li. ✉e-mail: hli@buct.edu.cn; khwu@iphy.ac.cn; lchen@iphy.ac.cn

mechanism, such as a nonlinear process, can lead to chiral symmetry breaking of two mirrored objects and give a competitive chirality—for example, in primordial chiral symmetry breaking[31]. It is therefore important to explore possible chiral recognition, characterization, transition, and amplification processes in achiral systems to understand the fundamental mechanisms of homochirality generation.

Here, we report on a chiral interface system constructed by the saturated adsorption of achiral carbon monoxide molecules on an achiral Au(111) surface. Combining STM with DFT calculations, we found the CO adlayer on Au(111) forms two kinds of regular chiral arrangements of heptamers, which are mirror-symmetry and non-superimposable with each other. When a high bias voltage is applied to the CO adlayer on Au(111), the chiral heptamers transform to a uniform phase consisting of CO monomers. After the bias voltage is lowered, the system experiences an auto-amplification during the recondensation of a chiral heptamer phase driven by the free energy of domain boundaries. Employing DFT calculations, we show that the asymmetry amplification is both kinetically feasible and thermodynamically favorable. Therefore, the adsorption behaviors of CO/Au(111) evidence that achiral molecules adsorbed on achiral substrates can form chiral-symmetrical structures, and the auto-amplification of the enantiomeric excess can be driven by domain boundaries. This finding reveals a chirality and homochirality from a pure achiral system, and provides a potential platform for

understanding heterogeneous asymmetric catalysis and the origin of homochirality through surface adsorption.

## Results

### Chiral structure of CO molecules adsorbed on Au(111)

Figure 1a is an STM image of a saturated CO monolayer adsorbed on an Au(111) surface. The unaffected herringbone reconstruction of the Au(111) surface can be discerned, in which the alternating wide and narrow flat regions are known as fcc and hcp parts due to different stacking sequence of top layer of Au atoms with respect to the underlayer, implying a weak interaction between CO molecules and the Au substrate. According to the orientations of CO arrangements with respect to the highest-symmetry crystalline direction of Au(111), there are two types of domains, A and B, as shown in Fig. 1b, and the distributions of domains are random and independent of the underlying herringbone orientations. It should be noticed that the long-range order of CO adlayer with the same orientation is not very perfect, and some lateral shifts often occur as shown in Supplementary Fig. 1. Figure 1c shows the fast Fourier transform (FFT) of Fig. 1b. For clarity, the main reciprocal spots are extracted, as shown in Fig. 1d. There are two sets of reciprocal spots (lattice vectors $q_1$ and $q_2$) from CO adlayer, corresponding to the lattice constants $a_1 = 3.8 \pm 0.1\,\text{Å}$ and $a_2 = 9.8 \pm 0.1\,\text{Å}$, respectively. To construct adsorption model, Au(111)–1 × 1 points (lattice vector $q_0$) can be derived from the points of herringbone reconstruction

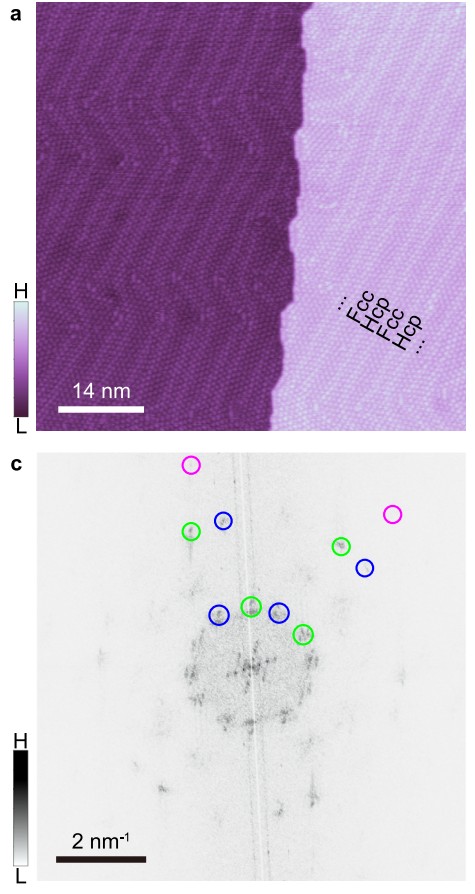

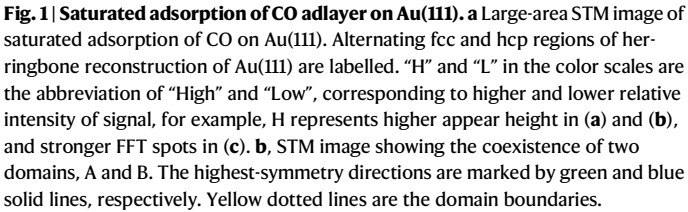

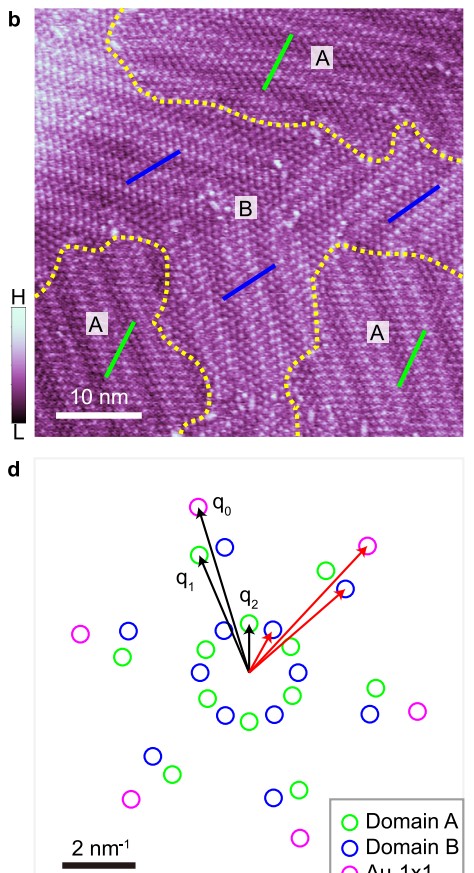

**Fig. 1 | Saturated adsorption of CO adlayer on Au(111). a** Large-area STM image of saturated adsorption of CO on Au(111). Alternating fcc and hcp regions of herringbone reconstruction of Au(111) are labelled. "H" and "L" in the color scales are the abbreviation of "High" and "Low", corresponding to higher and lower relative intensity of signal, for example, H represents higher appear height in (**a**) and (**b**), and stronger FFT spots in (**c**). **b**, STM image showing the coexistence of two domains, A and B. The highest-symmetry directions are marked by green and blue solid lines, respectively. Yellow dotted lines are the domain boundaries.

**c**, Corresponding 2D-FFT image of (**b**). **d** Schematic diagram showing the main reciprocal spots in (**c**) more clearly. The main spots are shown with green (domain A) and blue (domain B) hollow circles. The pink circles designate the spots of Au-1×1 derived from the herringbone pattern. The lattice vectors of Au, CO molecules, and heptamer arrangements are labelled $q_0$, $q_1$, and $q_2$, respectively. For clarity, black and red arrows are used to indicate the mirror relationships of domain A and B. The scanning parameters for all images are $V_{\text{tip}} = 40\,\text{mV}$ and $I = 5\,\text{pA}$.

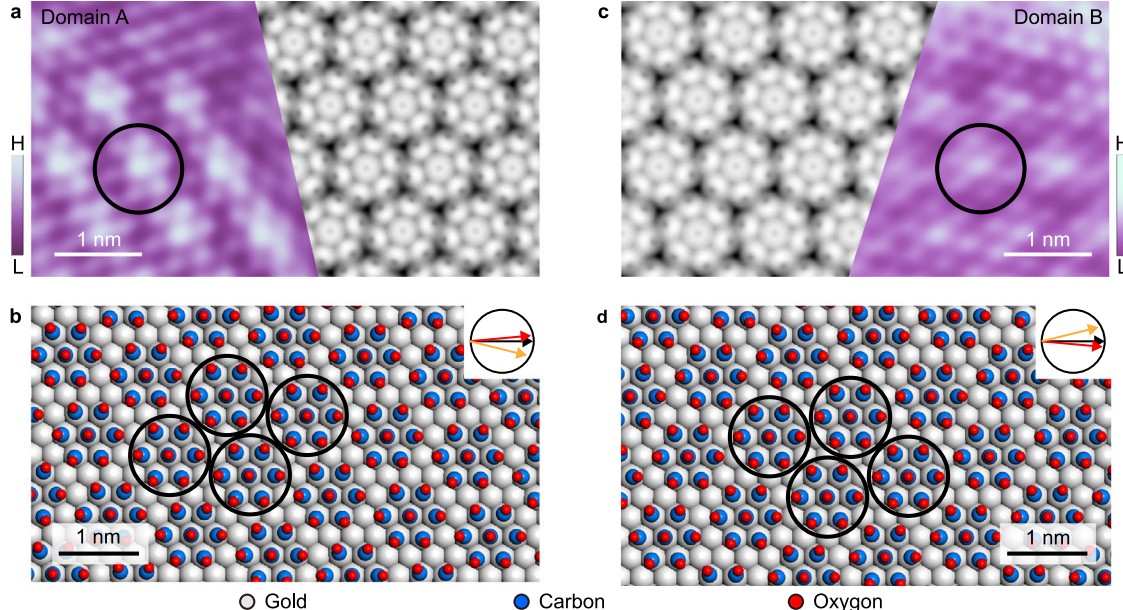

**Fig. 2 | The structure of CO chiral domains on Au(111). a** STM image (left panel) and simulated image (right panel) for domain A. **b**, The corresponding structural model for domain A. **c**, STM image and simulated image for domain B, **d**. The corresponding structural model for domain B. The STM images were acquired at $V_{tip}$ = 700 mV (**a**) and 500 mV (**c**), with $I$ = 5 pA. "H" and "L" in the color scales are the abbreviation of "High" and "Low". Gray, blue, and red balls in (**b**) and (**d**) represent Au, C, and O atoms, respectively. Black circles mark the heptamer building blocks. The inserts show the relative orientations of the highest-symmetry direction of Au-1×1 (black arrows), CO molecules (red arrows), and heptamer clusters (orange arrows) arrangements.

in the center of FFT image based on the geometrical relationship of 22 × √3 model. One can observe two groups of reciprocal spots in each series with different periodicities, which correspond to the two kinds of CO domains with different orientations (black arrows and red arrows). The lattice vectors of the two groups of reciprocal spots ($q_1$ and $q_2$) are rotated through the same angle clockwise and anticlockwise with respect to the Au(111) 1×1 lattice ($q_0$), suggesting that the two kinds of CO domains are mirror images of each other. The little distortion in Fig. 1c is due to thermal drift during the scanning process.

High resolution STM images of CO domain A and B are shown in Figs. 2a and 2c, respectively. Each bright protrusion corresponds to one CO molecule, which seems to follow a hexagonal arrangement. However, a detailed examination reveals that the CO molecules in both domains aggregate into a regular arrangement of 7-unit clusters (heptamers). The distance between neighboring heptamers is approximately 9.8 Å, corresponding to the inner reciprocal spots ($q_2$) in Figs. 1c and 1d. The distance between neighboring CO molecules is approximately 3.8 Å, corresponding to the middle reciprocal spots ($q_1$). Accordingly, the two types of CO heptamer domains can be denoted as (√13 × √13)R13.9° with respect to Au(111) 1 × 1 lattice for domain A, and (√13 × √13)R-13.9° for domain B. Additionally, the domains of CO cluster phases with two kind of orientations can extend dozens of nanometers, and cross several fcc and hcp regions as well as even elbow areas. Therefore, we assume the influence from different regions of herringbone on the formation of cluster phases is negligible.

To check the accuracy of the above structural model, we performed DFT calculations on the saturated adsorption structure of CO molecules on Au(111): the fully relaxed results are shown in Figs. 2b and 2d. It is found that the CO molecules spontaneously aggregate into the clustered configuration, in which the central molecule sits atop an Au atom site, while the six surrounding CO molecules are located on the adjacent top sites, forming a close-packed heptamer. In addition, due to repulsion between the dipole moments of adsorbed CO molecules, the oxygen atoms in surrounding CO molecules are slightly pushed away from each other, leading to an inclined adsorption conformation. The insets in Figs. 2b and 2d show the orientations of the highest-

symmetry directions of Au(111)−1 × 1 (black arrows), CO molecules (red arrows), and heptamers arrangements (orange arrows). The orientation of CO molecules and heptamers in domain A are rotated, respectively, by 5.2° anticlockwise and 13.9° clockwise relative to the Au(111)−1 × 1 direction. In domain B, CO and heptamers are respectively rotated by 5.2° clockwise and 13.9° anticlockwise, demonstrating that domain B is mirror-symmetric to domain A. Further, domains A and B are non-superimposable by any translation or rotation, and can thus be called the left- and right-handed cluster phases. Additionally, the simulated STM images of domains A and B based on the partial charge density agree perfectly with the experimental STM images (Figs. 2a and 2c), which validates the present structural models.

## Phase transition induced by bias voltage
When the bias voltage exceeded 1.0 V during scanning, we found that the clustered phase gradually faded (Figs. 3a and 3b) and disappeared at 3.0 V, where the CO molecules exhibit a close-packed structure (Fig. 3c). Correspondingly, the FFT images (Fig. 3d-f) indicate that the inner reciprocal spots ($q_2$) related to the (√13 × √13)R ± 13.9° structures vanished, leaving the middle reciprocal spots with lattice vector $q_1$ unchanged. This means that the orientation of CO molecules with respect to the substrate is not changed, although the cluster configuration is disrupted. The lower contrast of STM image at high bias stem from the metastable phase of the CO monomers, which will be easy to diffuse on the surface, causing the blurred feature on some area of surface. Further FFT images under different bias are shown in Supplementary Fig. 2. Not only the CO monomer but the refined structure of herringbone is clearly observed, which can rule out the loss of resolution of the tip under high bias. Therefore, we built the structure models of CO monomers (named the uniform phase) as shown in Figs. 3g and 3h, revealing the rearrangement of CO monomer at nearby positions after disaggregating the heptamer clusters due to the higher diffusion ability.

The adsorption-free energies ($\Delta G$) of both the cluster and the uniform phase structures derived from the DFT calculations are listed in Table 1, which are −0.43 eV and −0.38 eV for the cluster and uniform

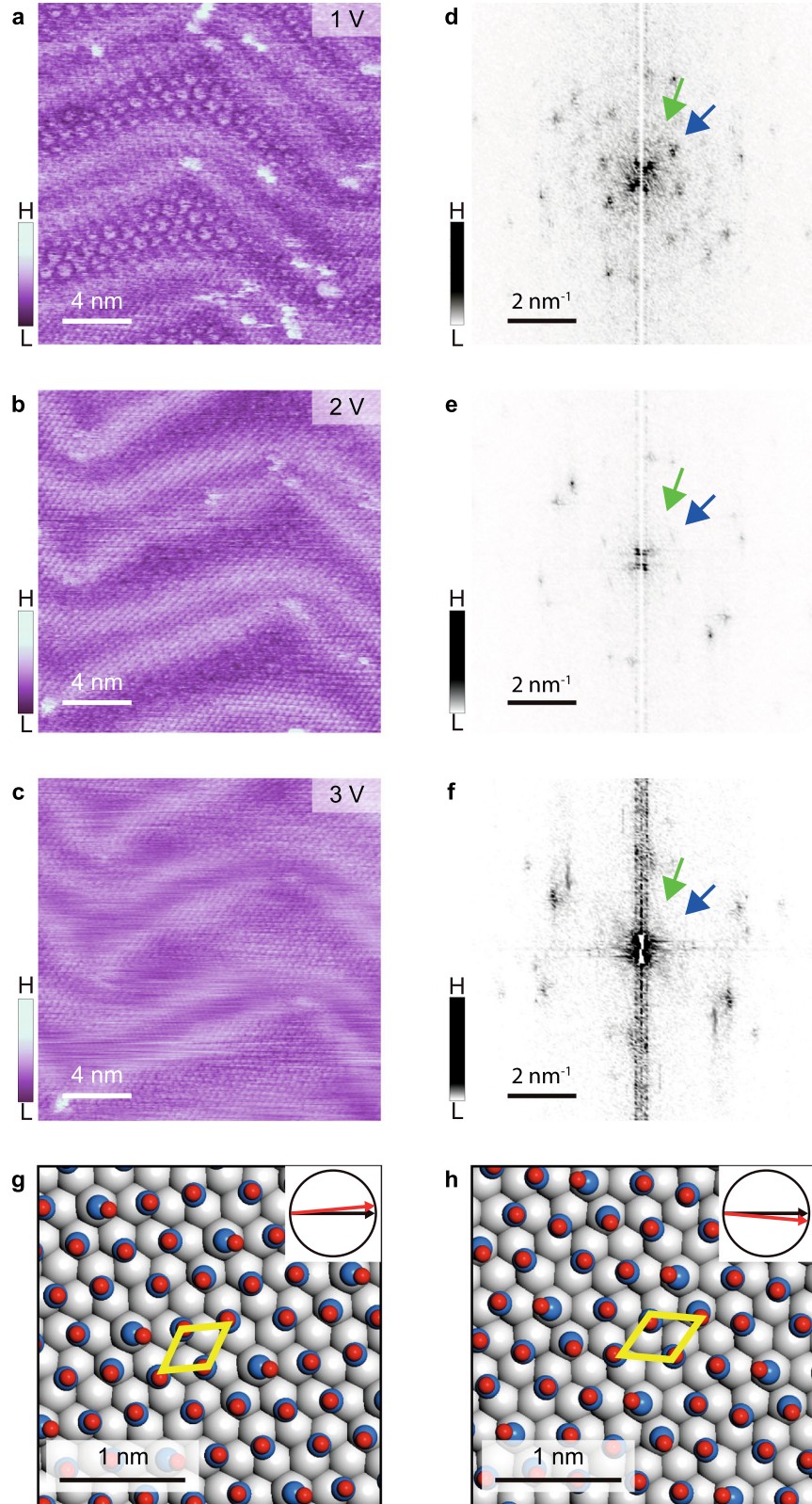

**Fig. 3 | Phase transition of CO clusters under an electric field. a–c** STM images of the same area of a CO adlayer on Au(111) taken continuously at tip biases of 1, 2, and 3 V, respectively. **d–f** Corresponding FFT images of panels **(a-c)**. Green and blue arrows marked the spots of left- and right-handed cluster phases. "H" and "L" in the color scales are the abbreviation of "High" and "Low". The tunneling currents are 5 pA. **g–h** Possible structural models of left-rotated **(g)** and right-rotated **(h)** domains of uniform phase consisting of CO monomers. Gray, blue, and red balls represent Au, C, and O atoms, respectively. Yellow rhombuses represent the 1×1 lattice of uniform phases. The corresponding orientation relationships between Au-1×1 (black arrows) and CO monomer (red arrows) are shown in the insert diagrams.

**Table 1 | Adsorption free energy ΔG (T = 5 K) of different adsorption structures derived from DFT (unit: eV)**

| Structure model | | E-field (eV Å⁻¹) | | | |
|---|---|---|---|---|---|
| | | 0.00 | 0.10 | 0.20 | 0.35 |
| Cluster | Left-handed | −0.432 | −0.430 | −0.428 | −0.426 |
| phase | Right-handed | −0.432 | −0.430 | −0.429 | −0.426 |
| Uniform | Left-rotated | −0.379 | −0.383 | −0.386 | −0.390 |
| phase | Right-rotated | −0.379 | −0.389 | −0.392 | −0.396 |

phases, respectively. We further calculated the adsorption free energies under the electric fields induced by the STM tip. The adsorption structures are optimized, while the adsorption energies are derived by setting **E**-field at 0.10 eV Å⁻¹, 0.20 eV Å⁻¹, and 0.35 eV Å⁻¹, respectively. As listed in Table 1, ΔG of the cluster phase is gradually reduced with the increase of the electric field. By contrast, ΔG of the uniform phase shows the increasing trend with the field, suggesting the relative stabilities of the cluster phase and the uniform phase are lowered and enhanced at high electric field, respectively. Therefore, the DFT calculations confirm that the transition between the cluster and uniform phases could be induced by the high electric field from the STM tip.

Furthermore, we estimate the minimum diffusion barrier of CO monomer, CO in trimer and heptamer on Au(111), as shown in Supplementary Fig. 3. Compared with the diffusion barrier of single CO and CO in trimer (0.06 eV), the diffusion barrier of CO in heptamer is higher at 0.152 eV, indicating that the heptamer is more stable than a monomer and trimmer on substrate. This barrier is less than that of CO molecules on other noble metal surface[32], supporting an easier diffusion for a moderate tip voltage. In addition, the entire diffusion barrier in the transition from left-handed cluster phase to the right-handed cluster phase is estimated to be 1.39 eV (Supplementary Fig. 4d), consistent to the diffusion barrier of each CO molecule shown in Supplementary Fig. 3h. It is noteworthy that the diffusion barrier can be apparently reduced by the electric field (see Supplementary Fig. 5-7). This indicates that the electric field induced by the STM tips can promote the motion of CO molecules, making the phase transition easier to occur.

### Chiral recondensation and chiral amplification

As the STM bias was decreased below 1.0 V, the uniform phase spontaneously recondensed back to the more stable cluster phase. Figure 4a-d show a series of STM images taken of the same area of CO adlayer on Au(111). Initially (−1.0 V, Fig. 4a), the left- and right-handed cluster phases spontaneously formed and coexist on the surface due to stable adsorption energies. When the bias is increased to +1.5 V, the cluster phases fade away in some areas (hcp regions) (Fig. 4b). When the bias is increased further to +3.5 V (Fig. 4c), the CO adlayer on the whole surface transitions into a uniform monomer phase. When the bias is decreased back to +1.5 V, the cluster phase occurs preferentially in the fcc regions of the surface (Fig. 4d). Significantly, the left-handed phase was completely suppressed, whereas the right-handed cluster phase prevailed. Figures 4e and 4f are the STM image and the corresponding FFT image obtained at −1.0 V in the end. It is evident that the whole surface is covered by the right-handed cluster phase, same for an opposite bias polarity (−3.5 V in Supplementary Fig. 8). Thus, we infer that such recondensation process from uniform CO monomers to the cluster phase is accompanied by an asymmetric amplification of chirality.

## Discussion

Saturated adsorption of achiral CO molecules on various substrates have been observed to form stable phases consisting of clusters due to spontaneous condensation at low temperature. For examples, hexamer and heptamer structures have been suggested on the (111) plane of bulk α-CO[33–35], as well as a CO adlayer on graphite[36], Ag(111)[37] and

Cu(111)[38]. Other chiral systems of clusters induced by symmetry-breaking adsorption process have also been reported, such as water adsorption on Ag(111)[39] and NaCl(001)[40]. In our experiment, CO molecules on Au(111) form different regular adlayer with an organized chirality, and it is significant that the chiral CO-Au system can experience an amplification of enantiomeric excess induced by recondensation from the metastable uniform phase.

By applying a high bias voltage, adsorbed CO molecules can easily overcome the barrier and diffuse on the surface, leading to the dissociation and recombination of CO clusters. As a result, the whole system is transitioned into a metastable state (uniform phase) from the stable racemic structure (cluster phase), providing the preliminary environment for further chiral excess and amplification. The threshold voltages for the transition of the chiral cluster domain to a uniform phase varies as the tunneling current is summarized in Supplementary Fig. 9, suggesting the electric field under positive bias and electric or vibration excitation under negative bias could induce phase transition in a CO/Au system.

After lowering the high voltage, the metastable uniform phase is recondensed into cluster phases. During this recondensation process, the statistic noise fluctuation will break the chiral symmetry and induce the spatial segregated domains, which result in the formation of enantiomeric excess (ee)[41]. As reported before[42,43], on the reconstructed Au(111) surface the CO adlayer has different interaction strengths with the hcp and fcc regions. Due to the very low diffusion rate of CO at low temperature, CO molecules on various fcc regions form separated small domains. These domains with stochastic chirality because of equal energy could extend to the hcp regions, and finally construct a saturated CO cluster phases on the whole surface. It should be noticed that the chirality of the small cluster domain is statistically random, which determines that the chirality of the domain after amplification is also random (left- or right-handed), as shown in Supplementary Fig. 10. Combined with the almost unanimously apparent heights of left- and right-hand cluster phases, the possibility of chiral transfer from tip chirality to induce ee[44–47] can also be ruled out.

The domain with homochirality will be grown further by successive operation process at high bias voltage, revealing the chiral amplification. The driving force of the chiral amplification is the free energy of domain boundaries, which are formed between adjacent domains with different chirality due to the mismatched orientations. The distribution of the boundaries is random at first, as shown in STM images (Fig. 1b, Supplementary Fig. 10), which accumulates edge-free energy and makes the whole system metastable. To minimize the free energy of the whole system, the domain boundaries prefer to become shorter and straighter after phase transition, which induce the growth of the domain with homochirality. As an example, a group of STM images recording successive evolution of domain boundaries between two coexisting domains of CO cluster phases as shown in Supplementary Fig. 11. Figure 5 illustrates the operation process containing multiple cycles of phase transition on a larger area (50×50 nm²) of CO cluster phases, and each STM image represent the change of cluster phases after one operation cycle at high voltage. The size of domain B shrinks gradually, and finally the whole area turns into one domain A with homochirality, suggesting chiral amplification to occur. Additionally, the successive amplification process of homochiral domain B is shown in Supplementary Fig. 12.

The "boundary-driven chiral amplification" is supported by the modified Frank model based on statistical fluctuation, which simulated the chiral separation under different local random fluctuations induced by reaction-noise[41,48]. The original Frank model constructed by F. C. Frank[49] was deterministic, expressed by a simple system consisting of two chiral autocatalytic reactions between achiral and chiral (L and D enantiomers) reagents (with a rate constant of $k_1$), coupled with a terminal competition reaction between them that produce an

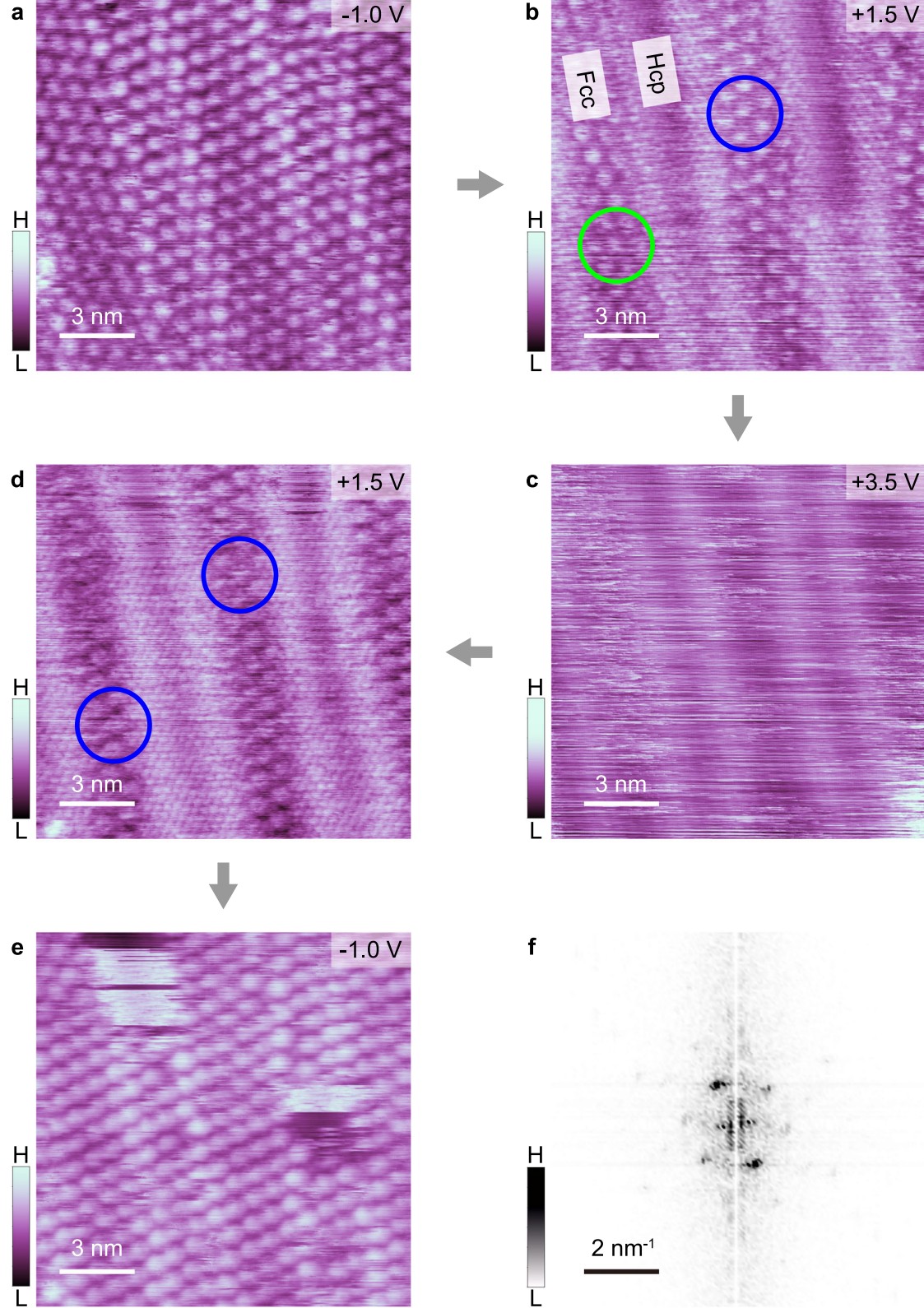

**Fig. 4 | Asymmetry amplification of chirality in CO clusters on Au(111). a–d** STM images obtained by successively scanning the same area of a CO adlayer on Au(111) with, respectively, −1.0 V, + 1.5 V, + 3.5 V, and then back to +1.5 V. Green and blue circles marked the spots of left- and right-handed cluster phases. "H" and "L" in the color scales are the abbreviation of "High" and "Low". **e, f** STM image and the corresponding FFT image obtained at −1.0 V for the same area as covered by panels (**a-d**). These show a homochiral cluster phase after decreasing the electric field. The green and blue circles represent, respectively, left- and right-handed domains. Scanning currents are 5 pA for all images.

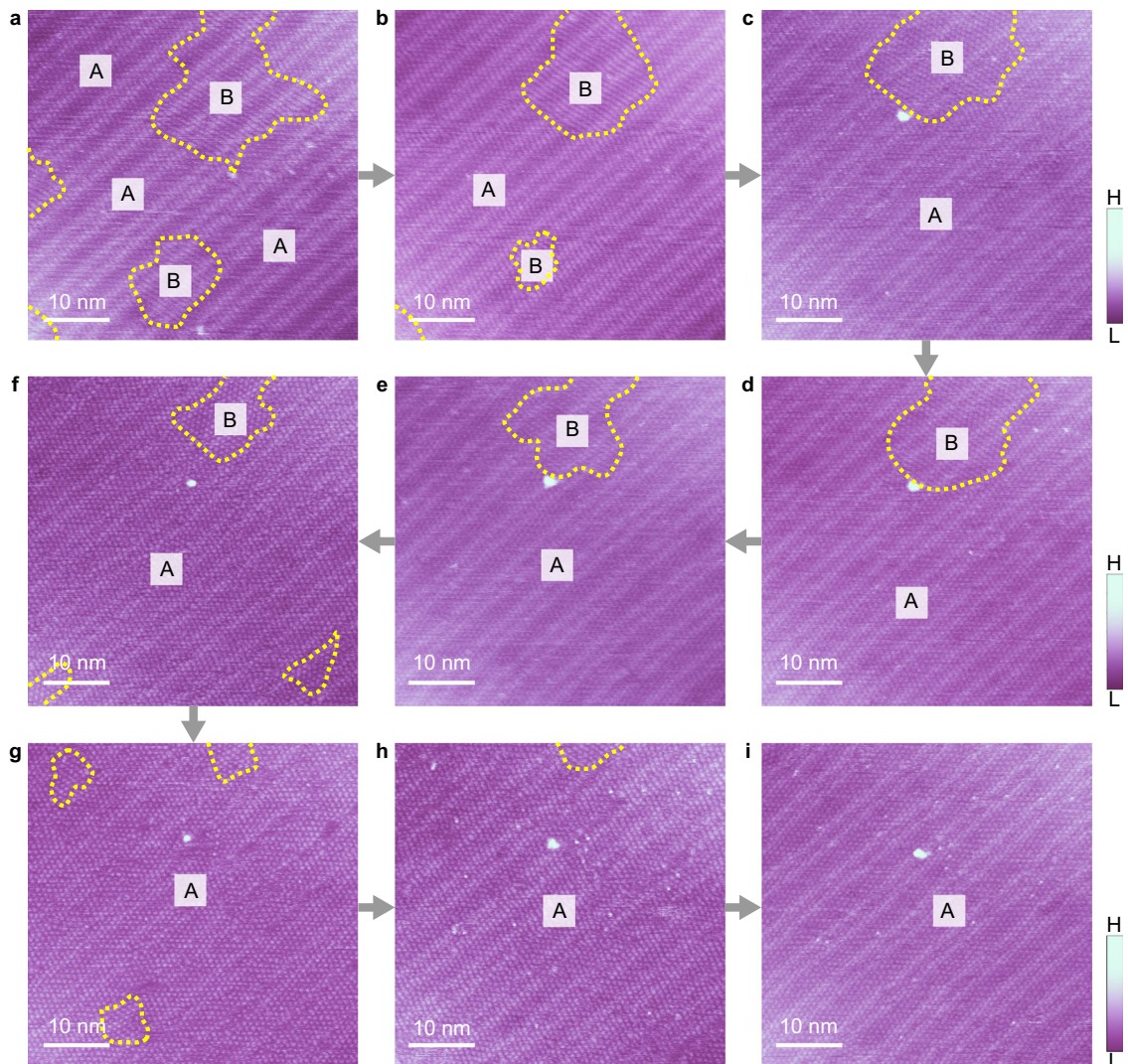

**Fig. 5 | The evolution of the homochirality of CO cluster phase on Au(111). a–i** Each STM image is scanned at the low bias image (−1.0 V, 6 pA) after one-time scanning at high bias (−3.5 V, 6 pA). Gray arrows represent the operation sequence. Yellow dotted lines marked the domain boundaries.

achiral inhibition product (with a rate constant of $k_2$):

$$\frac{dn_L}{dt} = (k_1 - k_2 n_D)n_L \qquad (1)$$

$$\frac{dn_D}{dt} = (k_1 - k_2 n_L)n_D \qquad (2)$$

where $n_L$ ($n_D$) is the concentrations of L (D) enantiomer. Through numeric calculation, it is given that:

$$ee(t) \equiv n_L - n_D = (n_{L0} - n_{D0})e^{k_1 t} \qquad (3)$$

The evolution of $ee(t)$ under original perturbation $ee(0)$ demonstrates that the final configuration of the system strictly depends on its initial excess.

Systems in nature are subject to various disturbances at any given time. The Frank model can be modified to account for stochastic fluctions, which cause the system to deviate slightly from its equilibrium states[50]. Over time, two enantiomers may become equal in excess, or there may be a possibility for the enantiomer with an enantiomeric deficiency to become the dominant final state. We can define $P_w$ as the probability of the enantiomer $w$ (L or D) to becoming

the final homochiral state

$$\frac{P_D}{P_L} = \lim_{t \to \infty} \frac{1}{M}\sum_{i=1}^{M}\frac{n_D(t)}{n_L(t)} = e^{-\alpha ee}. \qquad (4)$$

Here, $M$ is the number of different stochastic trajectories and $\alpha$ is the coefficient depending on the specific system. Since $P_L + P_D = 1$,

$$P_D \equiv (1 + e^{\alpha ee})^{-1}. \qquad (5)$$

As a result, the probability of forming homochiral states $P_D$ decreases exponentially with any initial enantiomeric excess mediated by statistic fluctuation, highlighting the role of statistical fluctuations in the emergence of homochirality (see Supplementary Fig. 13).

A current frontier research area is asymmetric catalysis, which may provide some clues for us to understand the process of the chiral amplification of small ee although with distinct mechanisms of the formations of ee. A chiral compound with single enantiomers can be synthesized from simple achiral materials using organic small molecules as catalysis, reflecting a single chiral collapse process from degenerate or nearly-degenerate chiral enantiomers. Despite the essential differences between molecular chirality and adsorption chirality, there may be some intrinsic commonality in the chiral

condensation and chiral separation mechanisms. Our system is based on the chiral structures constructed by the adsorption of the simplest polar molecule on an achiral transition metal surface, which may provide a potential research platform for the investigation of chiral condensation or chiral phase transition.

## Methods

### Preparation and characterization of sample
Our experiments were carried out in a home-built low-temperature STM/molecular beam epitaxy system with a basic pressure of $2 \times 10^{-11}$ torr. The single-crystal Au(111) substrate was cleaned by cycles of argon ion sputtering ($5 \times 10^{-4}$ Pa, 1 kV) and annealing at about 700 K. The CO adlayer was prepared by dosing CO gas (purity: 99.9%) on Au(111) at liquid-helium temperature in a UHV chamber with a pressure of $2 \times 10^{-10}$ torr for more than 1 h. The STM measurements were taken at 5 K. All the STM data were analyzed and rendered using WSxM software[51].

### Successive phase transition and chiral amplification
In our experiments, two kinds of methods were used to stimulate and check the different surface states before and after phase transition. First, the high bias voltage is applied through scanning the area of CO adlayer. The typical scanning time for one image is 5-7 min, corresponding to 7-8 ms per pixel, which is long enough to complete the phase transition. In one operation process or one cycle, we scanned an area with CO cluster phases at low voltage firstly. After one image is obtained, we set the bias to a higher value, corresponding to the high field on, and scanned the same area to induce the cluster phases transition to uniform phase. When the image at higher bias is obtained, the bias is set to the lower value, corresponding to the high field off, and we rescanned the same area to check the result after phase transition. Such operation process is performed on the whole scanning area. Second is local voltage pulse applied at a fixed position under the tip. The similar STM images at low bias voltage were obtained, but the higher bias is applied by tip pulses at single-point rather than scanning the entire area (see Supplementary Fig. 14). And we declare that all observations were measured repeatedly well in our STM experiments.

### Theoretical calculations
The DFT calculations were carried out with Perdew, Burke, and Ernzerhof (PBE) functional[52] in Vienna Ab Initio Simulation Package (VASP)[53]. The projector augmented-wave (PAW) pseudopotential[54,55] was employed for treating the atomic core electrons, and the plane-wave cutoff energy was set to 450 eV. In addition, the Van der Waals (vdW) correction was included with the Grimme PBE-D3[56,57] method for deriving the more accurate CO adsorption energy, more details can be seen in the Supplementary Table 1. Simulated STM topographic images of domain taken in an energy range of −1 to 1 eV. When considering the electric field, we optimized the adsorption structures and derived the energies by setting **E**-field = 0.00 V Å⁻¹, 0.10 V Å⁻¹, 0.20 V Å⁻¹, and 0.35 V Å⁻¹, respectively.

In the simulation models of adsorption structures, four layers of Au atoms were employed to mimic the Au(111) surface. The substrate was simulated by a hexagonal supercell with ($\sqrt{13} \times \sqrt{13}$) R ± 13.9° in-plane supercells (containing 7 adsorbed CO molecules). A $2 \times 2 \times 1$ Monkhorst-Pack $k$-mesh was used to sample the first 2D Brillouin zone. During the geometry optimization, the two uppermost layers and the CO molecules were relaxed and the lower two layers of Au atoms were fixed, and a vacuum region of ≥ 15.0 Å was used to exclude periodic surface–surface interactions. The CO molecular films were initially constructed according to the STM images, where the CO molecules are placed vertically and about 3.0 Å above the Au (111) surface, and the intermolecular distance between CO molecules was in the range of 3.7 to 4.0 Å. All the adsorption systems were fully relaxed until reaching force criterion of <0.02 eV Å⁻¹.

Based on vibrational frequency analysis[58], the zero-point energy (ZPE) correction and entropy ($\Delta S$) were also calculated to derive the adsorption free energy ($\Delta G$) through

$$\Delta ZPE = ZPE_{nCO|Au111} - ZPE_{Au111} - n \times ZPE_{CO(g)} \qquad (6)$$

$$\Delta S = S_{nCO|Au111} - S_{Au111} - n \times S_{CO(g)} \qquad (7)$$

$$\Delta E_{ads} = E_{nCO|Au111} - E_{Au111} - n \times E_{CO(g)} \qquad (8)$$

$$\Delta G = (\Delta E_{ads} + \Delta ZPE - T\Delta S)/n \qquad (9)$$

where $\Delta E_{ads}$ is the adsorption energy of CO molecules on Au(111) surface, and $\Delta ZPE$ and $\Delta S$ are respectively the differences of the zero-point energy and the entropy of the system before and after adsorption, $n$ is the number of CO molecules on Au(111) surface.

## Reporting summary
Further information on research design is available in the Nature Portfolio Reporting Summary linked to this article.

## Data availability
The data that support the findings of this paper are available from the corresponding authors upon request.

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

## Acknowledgements

L.C. acknowledges the financial supports by Ministry of Science and Technology (MOST) of China (Grant No. 2018YFE0202700), Natural Science Foundation of China (NSFC) (Grant Nos. 12134019), and Chinese Academy of Sciences (Grant No. XDB30000000, YSBR-054). H.L. acknowledges the financial supports by NSFC (21935001) and Beijing Municipal Natural Science Foundation (Grant No. Z210016). K.W. acknowledges the financial supports by MOST of China (2021YFA1400500) and NSFC (11825405, 11974322).

## Author contributions

L.C. and K.W. designed and conceived this research. H.Liu and Y.H. performed the experiments under the supervision of L.C. and K.W., Heping Li performed the calculation works under the supervision of Hui

Li, B.F., Y.Q.Z., and P.C. participated in the data analysis and discussion. All authors contributed to the writing of the manuscript.

## Competing interests
The authors declare no competing interests.
