## [Peer Review File · Nature Communications]

Condensation and asymmetric amplification of chirality in achiral molecules adsorbed on an achiral surfaceReviewers' Comments:

Reviewer #1:

Remarks to the Author:

Title: Condensation and asymmetric amplification of chirality in achiral molecules on an achiral surface

Authors: H. Liu, ... L. Chen

Ms. No. : NCOMMS-22-33717

The authors have studied the low temperature adsorption and ordering of CO on the herringbone reconstruction of the Au(111) surface. Using STM, they have observed the long range ordered structures formed by CO at high coverages. In particular, they have observed a $(\sqrt{13}\times\sqrt{13})R\pm 13.9^\circ$ structure based on the formation of CO heptamers. This is a chiral phase with both enantiomer domains visible in a single $70 \times 70 \text{ nm}^2$ image.

The key discovery that the authors have studied is the conversion of structure during application of a local voltage using the STM bias ongoing from low bias to high bias and back to low. During this perturbation, the $(\sqrt{13}\times\sqrt{13})R\pm 13.9^\circ$ structure converts at high bias to a phase in which the heptamer clusters are disaggregated. As the bias is decreased, the $(\sqrt{13}\times\sqrt{13})R\pm 13.9^\circ$ heptamer cluster phase is reformed but in one enantiomeric domain across the entire image. The authors cite this as an example of autoamplification of enantiomeric excess.

The authors make a number of assertions that are interesting but in my opinion are not completely supported by the data, or are very hard to extract from the data that is shown. In addition, many parts of the manuscript need to be improved before considering this for publication.

1. My primary difficulty in reviewing this has been that the figures are too small to allow the reader to examine them in detail. In addition, the choice of colors used for symbols superimposed on the STM images have very poor contrast with the background color. The figures need to be printed as full page objects.

2. Figure 1. The dashed lines separating the domains in 1b are barely visible when printed. Also, how are the domain boundaries identified? I do not see the boundary drawn between A and B domains at the top of the figure. Both domains in that region seem to be the same. Conversely, there are clearly some domain boundaries crossing through the B domain at the center of Figure 1b. The FFT shown in Figure 1c has very poor image quality. The bigger issue is that I cannot see some of the features rendered in the schematic of Figure 1d (and, I can see some spots that are not in the rendering). Again some of the colors in Figure 1d have very poor contrast (when printed).

3. Figure 2. I am not sure what the point is for drawing lines between inequivalent CO molecules in the $(\sqrt{13}\times\sqrt{13})R\pm 13.9^\circ$ lattice. Also, in the printed version, it is almost impossible to see the individual features of the CO molecules in the schematic drawings.

4. Figure 3. What is the Woods notations for the uniform CO lattice. I do not see it in the caption and I could not find it in the text.

5. Some of these comments apply to the remaining figures.

6. Line 19. The proper term for 'chiral-symmetric' is 'disymmetric'.

7. Lines 58-61. This sentence is confusing.

8. Line 112. The authors start to discuss the bias voltage dependence of the STM images. Image contrast does depend on tip bias in STM. Is it possible that the disappearance of the $(\sqrt{13}\times\sqrt{13})R\pm 13.9^\circ$ overlayer has more to do with image contrast than physical change of the overlayer structure?

9. Line 122. The authors make a comparison of the distance between the adsorbed CO monomers and 'the average molecular distance in a pure CO gas phase'. This does not make sense. Separation in the gas phase is pressure dependent. The paper cited is a computational estimation of the CO-CO distance

in a (CO)₂ dimer, presumably in an in silico vacuum.

10. Line 125. The authors discuss the conversion of the CO monolayer to the homochiral ($\sqrt{13} \times \sqrt{13}$)R $\pm 13.9^\circ$ lattice and the fact that this occurs across the entire image but that it chooses randomly between chiral +13.9° and -13.9° orientations. How many times has this been observed? How well documented is it that this occurs randomly.

11. Line 125. When a homochiral domain has been formed does its homochirality extend beyond the edge of the image? If so, how far?

12. Line 152. The authors point out that the DFT derived adsorption energies of the two enantiomer domains are identical. There is nothing remarkable about this. In fact, when using DFT it would be impossible to observe otherwise.

13. Line 191-193. The authors state that the auto-amplification of chirality that they observed is similar to the asymmetric catalysis for which the Nobel Prize was recently awarded. However, the whole point of the amplification that they observe is that it occurs without a chiral bias. Asymmetric catalysis employs a chiral catalyst.

14. On that note there is work by ECH Sykes (~2005-2010) which demonstrates that STM tips can be chiral. This might be relevant to their observations.

Reviewer #2:

Remarks to the Author:

This work is quite interesting, in which the amplification of chiral domains (CO heptamers) by the action of an external electric field is discussed. This effect is studied in a system formed by achiral molecules and surfaces with chiral domains without an external electric field. The extent of these chiral domains depends on the strength of the electric field; the larger the field, the smaller the domains with a prevalence of CO monomers. When turning off the electric field, chiral domains are re-established, and new configurations are acquired. In this process of switching on/off of the electric field, the extension of the chiral domains change, and eventually, the system becomes homochiral for zero electric field. That is, the chiral symmetry breaking occurs when a slight enantiomeric excess, which appears due to thermodynamic fluctuations or a chiral perturbation, is amplified. This general effect is already well known, as discussed by Kondepudi in the 1990s, but perhaps not for systems similar to the one presented here. The experimental part is presumably complete, but the theoretical considerations are only basic. For example, neither quantum nor classical dynamics were given, nor the kinetic was modeled employing differential or stochastic equations.

In summary, this work finds a novel medium for heterogeneous asymmetric catalysis through surface adsorption. The result could be significant in understanding heterogeneous asymmetric catalysis via surface adsorption. This general effect is already well known, but perhaps not for systems similar to the one presented here. The work supports the conclusions and claims but needs more explanation on theoretical estimations. The experimental analysis is conclusive and only requires a revision in theoretical discussions. The methodology is acceptable and meets the expected standards in the field. However, it is necessary to show more details to reproduce the experimental and theoretical works.

I have some major issues as: It is necessary to

- 1) discuss better the possible origin of the enantiomeric excess and the amplification mechanisms;
- 2) give more details about DFT calculations concerning different initial conditions and electric fields. For example, how were extensive DFT calculations?
- 3) discuss that the value $0.007 \text{ eV} \sim 0.7 \text{ kJ/mol}$ is less than the usual numerical error;
- 4) estimate the effect of the electric field on the energy barriers;
- 5) describe the non-linear amplification processes, for example, the authors cited the Frank model, but there are no discussions and kinetic equations;
- 6) More details about the experiment, mainly concerning the process of turning the field on and off, time, and periodicity. The work apparently indicates that only one cycle was applied concerning the electric field, i.e., field off, field on, and field off. However, for the amplification of the enantiomeric excess, it would take more cycles. Another item that was not clear was the ramp of the electric field activation.

Some minor issues are:

- 1) The title must indicate the electric field;
- 2) In author contributions: Who is H.P. Li? Heping Li?
- 3) it is necessary to explain the notation of chiral symmetry phases $(\sqrt{13} \times \sqrt{13})R \pm 13.9^\circ$ adding primary references;
- 4) To show a better crystallographic description for fcc and hcp;
- 5) There are any formatting mistakes in the manuscript.

I am sending a brief discussion about the kinetic equations based on the Frank model to help to improve the work. Of course, this is only a guide to help, and I am not sure about the correctness of these models in describing the experiment. There is a pleiad of possibilities of models and theories. The results are from differential equations, but it is better to consider stochastic methods to include the fluctuations. The left figure below is the usual Frank model; the central figure is a

modified version of the Frank model. The final homochiral state is bounded following the experiment discussed here. As an observation, I do not know if this particular modified version is discussed in the literature. Finally, in the right figure, the modified Frank model was divided into two parts, one for the absence of the electric field and the other for the presence of the field. There are many possible descriptions, e.g., in the absence of the field, it is possible to use all the reactions of the modified Frank model. However, in the presence of the field, one can consider only the last step or other termination reactions. I have tested those, and the results are similar. It is possible to modify several parameters to describe the experiment better. (contains attached figure)

Reviewer #3:

Remarks to the Author:

The paper reports the observation of chiral superstructures in the saturated adsorption layer of CO molecules on Au(111). The electric field from an STM tip can switch the chirality of CO heptamer clusters and induce chirality transition of the racemic molecular domains. The behavior is explained by DFT calculations that compare the thermodynamic stabilities of the corresponding structures.

It is intriguing to find the chiral amplification phenomena in a simple system like CO/Au(111), which is a commonly studied system, nevertheless the evidence seems to be largely ignored. An in-depth understanding of the process may have general implications for the adsorption induced-chirality on surfaces.

I would like to ask the authors to clarify some issues that are not clear in the article.

1. More details of the DFT calculations should be provided. For example, the adsorption energies of hcp regions versus fcc regions and their electric field-dependency are absent. For a polar adsorbate, the effect of electric field polarity needs to be carefully discussed.
2. The FFT spots in Fig.1c and Fig.3d-f are hardly visible, and it appears that Fig.1c is distorted. How can it be so sure that the poor contrast in Fig.3c is not due to the loss of STM resolution at high bias?
3. I'm confused about "Although the left-handed and right-handed cluster phases can be formed on the whole Au(111) surface at low bias voltage". Are the chiral domains spontaneously formed or subjected to STM imaging?
4. It is oversimplified to calculate the diffusion barrier with the model of a CO trimer on a bare substrate. The molecules in the assembly have a coherent movement to switch their chirality. What is the threshold voltage for a chirality transition of domain? Is it possible to change the chirality of one or a few adjacent heptamer clusters in a racemic domain?
5. I would like to see a comparison of the original and after-electric field images of the same area to validate the proposed mechanism of domain boundary-driven amplification. The hcp and fcc regions should be clearly indicated. If only fcc regions can change their chirality under electric field, will the hcp regions remain intact after this perturbation?
6. I cannot understand the argument of "asymmetry amplification is both kinetically feasible and thermodynamically favorable". What is the kinetic factor in this case?

Reviewer #4:

Remarks to the Author:

Liu et al. report in the manuscript „Condensation and asymmetric amplification of chirality in achiral molecules on an achiral surface“ results obtained using scanning tunneling microscopy (at liquid He temperature) and density functional theory on the formation of chiral (mirror-symmetric) phases formed upon clustering or domain forming CO adsorbed on Au(111) – the latter surface apparently in its known Herringbone reconstruction.

The authors show by Fourier transforming STM micrographic images that long-range ordered areas or domains of CO clusters (6 or 7mers) are formed. The electric field imposed by the STM tip (using bias voltage of ca. 1 V) induces CO diffusion and the two observed domains (A and B or left- and right-handed) “dissolve” and re-assemble again, with in fact complete suppression of one of the phases (left-handed only, as described within lines 125 – 141). The authors also refer to the ExtendedData Fig. 1. The authors conclude on the so-called chiral amplification in this phase transition, meaning that one of the “chiral phases” dominates – a so-called enantiomeric excess occurs. DFT calculations using the Perdew, Burke, Ernzerhof generalized-gradient approximation to electronic exchange and correlation effects were employed to calculate adsorption energies (on DFT-level, i.e. total electronic energy differences only) of the two phases in comparison to a so-called uniform CO adsorption phase on the ideal, non-reconstructed Au(111) surface model.

Two main questions come to my mind after reading this manuscript:

- 1) How do the authors explain that only the left-handed or left-rotated form of the CO clusters are formed in their chiral amplification experiments. How many times have these experiments carried out

and are these experiments reproducible?

2) The Herringbone reconstruction of Au substrate is clearly visible on the STM images in Fig. 1. The CO adsorption – in my opinion- follows the $(22 \times \sqrt{3})$ reconstruction. To what extent – as the “thermodynamic window” or energy difference-window is narrow (see Table 1 and main text) – do the authors think that this reconstruction does NOT affect their observation. The reason I ask is the work by Hanke and Björk (PRB 87, 235422 (2013)), who showed that “neglect” of the reconstruction – especially the different straight and elbow areas involving different coordination numbers of Au atoms – affects adsorption energies.

3) I must criticize that the authors did not care about van der Waals type of dispersion effects.

4) As the experiments are carried out close to the absolute zero of temperature – T-induced effects are supposed to be small, but one should even then care about zero-point energy corrections. The DFT-modelling part is in this respect a bit too “coarse grained” given the very small energy differences the authors are aiming for to reproducibly describe and discuss.

5) The transition states for CO diffusion have been optimized using which methods? These aspects were not described in the manuscript. Is the diffusion energy barrier of 0.07 eV numerically converged? Is there really one imaginary mode along the diffusion path – I mean the barrier is close to the numerical error bars or uncertainties which are technically attainable ...

Overall, I cannot recommend this manuscript for publication in Nature Communications.

Reply to referee #1:

C: *The authors have studied the low temperature adsorption and ordering of CO on the herringbone reconstruction of the Au(111) surface. Using STM, they have observed the long range ordered structures formed by CO at high coverages. In particular, they have observed a $(\sqrt{13}\times\sqrt{13})R\pm 13.9^\circ$ structure based on the formation of CO heptamers. This is a chiral phase with both enantiomer domains visible in a single $70 \times 70 \text{ nm}^2$ image.*

The key discovery that the authors have studied is the conversion of structure during application of a local voltage using the STM bias ongoing from low bias to high bias and back to low. During this perturbation, the $(\sqrt{13}\times\sqrt{13})R\pm 13.9^\circ$ structure converts at high bias to a phase in which the heptamer clusters are disaggregated. As the bias is decreased, the $(\sqrt{13}\times\sqrt{13})R\pm 13.9^\circ$ heptamer cluster phase is reformed but in one enantiomeric domain across the entire image. The authors cite this as an example of autoamplification of enantiomeric excess.

The authors make a number of assertions that are interesting but in my opinion are not completely supported by the data, or are very hard to extract from the data that is shown. In addition, many parts of the manuscript need to be improved before considering this for publication.

R: Thanks for the referee's comments and suggestions. We carefully address all the questions in the following point-by-point.

C1: *1. My primary difficulty in reviewing this has been that the figures are too small to allow the reader to examine them in detail. In addition, the choice of colors used for symbols superimposed on the STM images have very poor contrast with the background color. The figures need to be printed as full page objects.*

R1: We are sorry for the inconvenience. We have uploaded the electronic-version figures with high resolution separately to the submission system for the detailed review. The colors for symbols in all figures in revised manuscript are also modified to enhance the contrast with the background color for better reading.

C2: *2. Figure 1. The dashed lines separating the domains in 1b are barely visible when printed. Also, how are the domain boundaries identified? I do not see the boundary drawn between A and B domains at the top of the figure. Both domains in that region seem to be the same. Conversely, there are clearly some domain boundaries crossing through the B domain at the center of Figure 1b.*

R2: Again, sorry for the inconvenience. The color of dashed lines marked the domain boundaries in revised Figure 1b has been changed for better distinguishment.

In our STM images, the two kinds of CO heptamers domains with different chirality (domain A and domain B) is identified as the area of heptamers with different orientations. Consequently, the domain boundaries are used to separate the area of heptamers with different orientations. To better show the position of domain boundaries, as an example, we superimpose the straight lines to reveal the orientations of the two neighbor domains in magnified upper part of Figure 1b, which has been added into the Supplementary Information as Supplementary Fig. 1a.

Moreover, we can notice the long-range order of CO heptamers with same orientation is not very good in STM images. The lateral shifts of CO heptamers often occur, but still keep the same orientation, which can be seen in the magnified central part of Figure 1b (see Supplementary Fig. 1b). In order to focus on the discussion of auto-amplification of enantiomeric excess, we regard the area of CO heptamers arrangements

with same orientation as one kind of domain and ignore the lateral shifts of heptamers inside one domain.

Supplementary Fig. 1: Enlarged images of Figure 1b. **a**, Difference in orientations of two kinds of the domains (domain A and B). Yellow dotted line marked the domain boundaries. Three equivalent directions of each domain are labeled by straight lines. Scale bar is 10 nm. **b**, Magnified central part of Figure 1b. Red points represent the CO cluster from right-hand domain (domain B). The yellow straight lines mark the lattice shifts usually occurred. Scale bar is 6 nm.

C3: *The FFT shown in Figure 1c has very poor image quality. The bigger issue is that I cannot see some of the features rendered in the schematic of Figure 1d (and, I can see some spots that are not in the rendering). Again some of the colors in Figure 1d have very poor contrast (when printed).*

R3: Sorry for the inconvenience. We try our best to adjust the contrast of Fig. 1c to enhance the image quality. The electronic-version figures with high-resolution are separately uploaded to submission system for better review.

To indicate what kind of features are rendered in the original Fig. 1d, we superimposed the circles on same positions in FFT image of Fig. 1c (see Fig. R1), which mark the most obvious features in FFT image. We agree that some reciprocal points corresponding to Au 1×1 lattice are quite dim compared with other features corresponding to CO heptamer arrangements and molecular arrangements due to the full coverage of CO molecules on Au surface, but the points corresponding to herringbone reconstruction are sharply clear in the center of image, and Au 1×1 points can be derived based on geometrical relationship of $22\times\sqrt{3}$ model. Even one or two Au 1×1 points still can be seen in Fig. 1c. To avoid misleading the readers, we only draw the circles to reveal the features corresponding to CO molecular arrangements (q_1), heptamer arrangements (q_2), and Au lattice within one quadrant in revised Fig. 1c. Additionally, some dim and weak spots are located between the q_1 and q_2 positions marked by black dotted circles in Fig. R1, which might be the secondary reciprocal points and are ignored to be marked in revised Fig. 1c.

Fig. R1 The detailed description of 2D-FFT image of revised Figure 1c. The main features include the contribution from domain A (green circles), domain B (blue circles) and Au atomic lattice (azure circle). Some weaker points marked by black dotted circles were ignored.

Furthermore, the color of Fig. 1d also has been changed for better contrast in the revised manuscript.

C4: 3. Figure 2. I am not sure what the point is for drawing lines between inequivalent CO molecules in the $(\sqrt{13}\times\sqrt{13})R\pm 13.9^\circ$ lattice. Also, in the printed version, it is almost impossible to see the individual features of the CO molecules in the schematic drawings.

R4: We thank the referee for the comment. The blue triangles in original Fig. 2b and 2d are supplemental signs to indicate the relative different arrangement of CO heptamers in unit cells of $(\sqrt{13}\times\sqrt{13})R\pm 13.9^\circ$ lattice, respectively. We agree that these triangles are not necessary, and have removed them from the revised figures to avoid the confusion.

Besides, we change the color of CO molecules in the schematic drawings of Fig. 2b and 2d in revised manuscript, and provide the electronic-version figures with high-resolution to submission system for better review.

C5: 4. Figure 3. What is the Woods notations for the uniform CO lattice. I do not see it in the caption and I could not find it in the text.

R5: Thanks for the referee's reminding. In principle, if considering the commensuration with Au(111) substrate, the Woods notations of uniform CO lattice should also be $(\sqrt{13}\times\sqrt{13})R\pm 13.9^\circ$ with respect to Au(111) 1×1 lattice due to same concentration as the cluster phase. Since the CO molecules are condensed into heptamers to form cluster phase, the arrangement of the heptamers can exhibit the reciprocal points corresponding to $(\sqrt{13}\times\sqrt{13})R\pm 13.9^\circ$ lattices in FFT images. However, because of the weaker interactions between CO molecules and Au(111) surface under higher tip bias, the FFT images only shows the reciprocal points corresponding to the 1×1 lattice of the uniform phase. Therefore, we didn't give the Woods notations for the uniform CO lattice in the main text. To avoid confusion, we removed the rhombus which indicate the $(\sqrt{13}\times\sqrt{13})R\pm 13.9^\circ$ lattice with respect to Au(111) 1×1 lattice, and added the rhombus indicating the 1×1 lattice of the uniform phase in revised Fig. 3g and 3h.

C6: 5. *Some of these comments apply to the remaining figures.*

R6: Sorry for the inconvenience. We have tried our best to improve the quality of images, and provided color scheme with high contrast. The electronic-version figures with high-resolution are separately uploaded to submission system for better review.

C7: 6. *Line 19. The proper term for 'chiral-symmetric' is 'disymmetric'.*

R7: Thank the referee. Referring to the referee's suggestion, the statement in the revised manuscript has been replaced by the sentences as following: "two disymmetric cluster phases consisting of CO heptamers with chirality are revealed."

C8: 7. *Lines 58-61. This sentence is confusing.*

R8: Sorry for unclear statements. We changed the sentence in the revised manuscript as follows: "Combing scanning tunneling microscopy (STM) with density functional theory (DFT) calculations, we found the CO adlayer on Au(111) forms two kinds of regular chiral arrangements of CO heptamers, which are mirror-symmetry and non-superimposable with each other."

C9: 8. *Line 112. The authors start to discuss the bias voltage dependence of the STM images. Image contrast does depend on tip bias in STM. Is it possible that the disappearance of the ($\sqrt{13}\times\sqrt{13}$) $R\pm 13.9^\circ$ overlayer has more to do with image contrast than physical change of the overlayer structure?*

R9: Thanks for the referee's comment. The bias-dependent features in STM image usually are mainly contributed from the electronic states of sample, which should still exhibit same translation symmetry at different energy due to the same structure. In some strong correlated electronic systems, the existence of charge density wave may induce the features in STM images with different translation symmetry. However, the CO molecule adlayer on Au(111) is stabilized by the weak van der Waals molecule-molecule and molecule-substrate interactions, which should not be a strong correlated electronic system. Furthermore, we can observe the clear features corresponding to big clusters and small monomers at low and high bias voltages respectively, which exhibit different translation symmetry in FFT results, indicating the origination of physical change of the CO overlayer structure at different bias voltages.

On the other hand, in STM image of Fig. 3c taken at high bias, we can still observe the obvious spots corresponding to CO monomers, which size is much smaller than CO heptamers observed in Fig. 3a taken at low bias. Therefore, the STM resolution is not loss at high bias. The lower contrast of STM image at high bias stem from the metastable uniform phase. The CO monomers in this phase will be easy to diffuse on the surface, causing the blurred feature on some area of surface. What's more, another group of FFT images at different bias is shown in Supplementary Fig. 2. Not only the CO monomer but the refined structure of herringbone is clearly observed, which can rule out the loss of resolution of the tip under high bias.

Supplementary Fig. 2: The evolution of bias-dependent FFT images obtained from the STM images taken on the same area of CO adlayer on Au(111). Scanning parameter: $V_{tip} = -0.1$ V for (a); 2.5 V for (b); 3.0 V for (c) with the same current 5 pA. Scale bars are 2 nm^{-1} .

Therefore, we believe the distinguished features in STM images and FFT results correspond to the phases transition of structures in CO adlayer. And the statements have been added into the revised manuscript (Line 127-132).

C10: 9. Line 122. *The authors make a comparison of the distance between the adsorbed CO monomers and 'the average molecular distance in a pure CO gas phase'. This does not make sense. Separation in the gas phase is pressure dependent. The paper cited is a computational estimation of the CO-CO distance in a (CO)₂ dimer, presumably in an in silico vacuum.*

R10: Thanks for the referee's comment. We agree this comparison may be not suitable. Therefore, we have deleted this statement in the revised manuscript.

C11: 10. Line 125. *The authors discuss the conversion of the CO monolayer to the homochiral $(\sqrt{13} \times \sqrt{13})R \pm 13.9^\circ$ lattice and the fact that this occurs across the entire image but that it chooses randomly between chiral $+13.9^\circ$ and -13.9° orientations. How many times has this been observed? How well documented is it that this occurs randomly.*

R11: We thank the referee for the question. The conversion of CO uniform phase to the cluster phases from high to low bias voltage will choose chiral $+13.9^\circ$ and -13.9° orientations randomly, which always occurs. To better illustrate this issue, we show a group of STM images recording successive phases transition on a larger area ($50 \times 50 \text{ nm}^2$) of CO cluster phase with several small domains (see Supplementary Fig. 3), in which each image is scanned at low bias voltage after one-time scanning at high voltage. One should notice that the domain A and domain B occur randomly on the scanning area after operation of phase transition at first. As the operation times increase, the size of domain increases and the density of boundary decreases. Unlike the small area in the main text ($15 \times 15 \text{ nm}^2$), more operation times are needed to achieve the homochirality on a larger area. As another example (see Fig. 5 in the revised manuscript), after several times of operation, the whole area become one domain of cluster phase with homochirality. Such observation is repeatable well in our STM experiments.

Supplementary Fig. 3: The evolution of domains of CO cluster phase on Au(111) pattern with successive operation cycles at high voltage. Each image is scanned at low bias (-1.0 V, 6 pA) after one-time scanning at high bias (+3.5 V, 6 pA). Gray arrows represent the operation sequence. Yellow dotted lines marked the domain boundaries. Scale bars are 10 nm.

Fig. 5 The evolution of the homochirality of CO cluster phase on Au(111). Each image is scanned at the low bias image (-1.0 V, 6 pA) after one-time scanning at high bias (-3.5 V, 6 pA). Gray arrows represent the operation

sequence. Yellow dotted lines marked the domain boundaries. Scale bars are 10 nm.

C12: 11. Line 125. *When a homochiral domain has been formed does its homochirality extend beyond the edge of the image? If so, how far?*

R12: We thank the referee for the question. In our experiments, the phase transition can be induced by the electron field between STM tip and sample. Usually, the radius of STM tip is in the range of 10 nm, but it's generally accepted that the atomic resolution arises from the atop atoms upon the tip apex, which is quite small (~1 nm). The effective electron field is confined in the vicinity of the top of STM tip during scanning (can less than 6-10 Å) [Sur. Sci. **282** (1993) 400-410; Sur. Sci. **429** (1999) 327-337]. Therefore, the homochiral domain formed from the phase transition is restricted well within the scanning area.

C13: 12. Line 152. *The authors point out that the DFT derived adsorption energies of the two enantiomer domains are identical. There is nothing remarkable about this. In fact, when using DFT it would be impossible to observe otherwise.*

R13: Thanks for the referee's comment. We agree that the fact of the identical adsorption energies of the two cluster phases is no remarkable. Therefore, we modified the sentence "The mirror-symmetric left-handed and right-handed cluster phases are energetically degenerate, and show identical adsorption energies obtained from the DFT calculations" to "The adsorption free energies (ΔG) of both the cluster and the uniform phase structures derived from the DFT calculations are listed in Table 1." in revised manuscript. (Page 7 Line 135)

C14: 13. Line 191-193. *The authors state that the auto-amplification of chirality that they observed is similar to the asymmetric catalysis for which the Nobel Prize was recently awarded. However, the whole point of the amplification that they observe is that it occurs without a chiral bias. Asymmetric catalysis employs a chiral catalyst.*

R14: Sorry for unclear discussion. The origin of homochirality is the spontaneous formation of a small enantiomeric excess (ee) followed by the amplification of the ee [Nat. Chem. 2015, **7**, 520-525; Top. Curr. Chem. **259**, 123-165 (2005)]. The formation mechanism of ee in different cases is distinct. The asymmetric catalysis needs an extra chiral catalyst to role as prochirality, while our result of local symmetry breaking is driven by statistical fluctuation. However, we want to express that it's the next-step amplification process in our work that might have some similarity to the asymmetric catalysis.

We revised the statements in manuscript (Line 272) as following: "A most frontier research area is "asymmetric catalysis", which may provide some clues for us to understand the process of chiral amplification of small ee although with distinct mechanisms of the formations of ee."

C15: 14. *On that note there is work by ECH Sykes (~2005-2010) which demonstrates that STM tips can be chiral. This might be relevant to their observations.*

R15: Thanks for the referee's suggestion. It is helpful for us to understand the origin of enantiomeric excess.

ECH Sykes's group has reported the observation of the intrinsic chirality in STM tips [Phys. Rev. Lett. **106**, 010801 (2011); Nat. Nanotech. 2011, **6**, 625; Chirality **24**: 1051-1054 (2012); Chem. Rec. 2014, **14**, 834]. In their work, they declared that 80% tip tested are chiral, which would produce different apparent height (tunneling probability) of two enantiomers and a large symmetry breaking (ee = 39 %) in electric excitation of molecule rotors. They speculated the chirality in tip results from geometric arrangement of the

atoms at the end of tip, similar as the chirality at the kink sites on bare metal. The tip chirality can be transferred to prochiral molecules in a coherent manner, inducing symmetry breaking and formation of enantiomeric excess.

In our work, we think the asymmetry amplification of chirality in CO clusters on Au(111) should not be from the chirality in tip. Firstly, the chirality in CO cluster phases originates from the integral adsorption situation of CO with respect to the Au(111) lattice. The apparent heights of left- and right-hand cluster phases doesn't show obvious difference, no matter how we change the tip state through pulsing or surface indentations. Secondly, under the same tip state (assuming there is a given chirality), the chirality of cluster phase transitioned from uniform phase is random, rather than a phase with preferred chirality.

The discussion above has been added into the revised manuscript (Line 222).

Reply to referee #2:

C: *This work is quite interesting, in which the amplification of chiral domains (CO heptamers) by the action of an external electric field is discussed. This effect is studied in a system formed by achiral molecules and surfaces with chiral domains without an external electric field. The extent of these chiral domains depends on the strength of the electric field; the larger the field, the smaller the domains with a prevalence of CO monomers. When turning off the electric field, chiral domains are re-established, and new configurations are acquired. In this process of switching on/off of the electric field, the extension of the chiral domains change, and eventually, the system becomes homochiral for zero electric field. That is, the chiral symmetry breaking occurs when a slight enantiomeric excess, which appears due to thermodynamic fluctuations or a chiral perturbation, is amplified. This general effect is already well known, as discussed by Kondepudi in the 1990s, but perhaps not for systems similar to the one presented here. The experimental part is presumably complete, but the theoretical considerations are only basic. For example, neither quantum nor classical dynamics were given, nor the kinetic was modeled employing differential or stochastic equations.*

In summary, this work finds a novel medium for heterogeneous asymmetric catalysis through surface adsorption. The result could be significant in understanding heterogeneous asymmetric catalysis via surface adsorption. This general effect is already well known, but perhaps not for systems similar to the one presented here. The work supports the conclusions and claims but needs more explanation on theoretical estimations. The experimental analysis is conclusive and only requires a revision in theoretical discussions. The methodology is acceptable and meets the expected standards in the field. However, it is necessary to show more details to reproduce the experimental and theoretical works.

R: Thank the referee's supporting and suggestions. We added more theoretical calculations and discussions in the revised manuscripts. All the questions will be addressed carefully in the following point-by-point.

C1: *I have some major issues as: It is necessary to (1) discuss better the possible origin of the enantiomeric excess and the amplification mechanisms;*

R1: Thanks for the referee's suggestion. We provided more discussions about the possible origin of the enantiomeric excess and the amplification mechanisms in revised manuscript.

In our system, the stable cluster phases of CO adlayer with chirality can transition to metastable uniform phase by high electric field. After withdrawing the high electric field, the metastable uniform phase is recondensed into cluster phases. During this recondensation process, the statistic noise fluctuation will break the chiral symmetry and induce the spatial segregated domains, which result in the formation of enantiomeric excess. The domain with homochirality will be grown further by successive operation process at high electric field, revealing the chiral amplification. The driving force of the chiral amplification is the free energy of domain boundaries. In order to minimize the free energy of whole system, the domain boundaries prefer to become shorter and straighter, which induce the growth of domain with homochirality. As an example, Supplementary Figure 4 contains a group of STM images recording successive operation of phase transition on a larger area of CO cluster phases. We can notice that the boundaries between domain A and B gradually become shorter and straighter, and the size of domain A became larger, reflecting the chiral amplification of CO cluster phase. Moreover, Roman Fasel *et al* also pointed out that boundaries between mirror domains are energetically unfavorable than that between domains with prochirality [Nature **439**, 449 (2006)]. The "boundary-driven chiral amplification" based on statistical fluctuation is consistent with the modified Frank model, and has been reported in [Nature **439**, 449 (2006); Chem. Phys. Lett. **431** (2006) 185], which

simulated the chiral separation under different local random fluctuations induced by reaction-noise.

Supplementary Fig. 4: The successive evolution of domain boundaries between two kinds of domains of CO cluster phase on Au(111). Each image is scanned at low bias (-0.1 V, 5 pA) after one-time scanning at high bias (-3.5 V, 5 pA). Gray arrows represent the operation sequence. Yellow dotted lines marked the domain boundaries. Scale bars are 8 nm.

Furthermore, the details of theoretical discussions will be shown in the following Reply to Comments.

C2: 2) give more details about DFT calculations concerning different initial conditions and electric fields. For example, how were extensive DFT calculations?

R2: Thanks for the referee's comment. We have added more details about the DFT calculations into the revised manuscript. It is noteworthy that we have not try many "different initial conditions" in the relaxation of CO adsorption structures, since the relaxation process for a random structure relaxing to the final ordered structure is far beyond our computational limit. Here, the initial adsorption positions of CO molecules were set according to the observed STM images, so that the final stable structures were relaxed in local minima not far from our initial structures. The initial height of the carbon atom in CO was set at ~3.0 angstrom above Au substrate.

First, according to the reviewer's suggestions, we added the following instruction of the computational details in the revised manuscript (Page 15, Methods: Theoretical calculations):

“The DFT calculations were carried out with Perdew, Burke, and Ernzerhof (PBE) functional [Phys. Rev. Lett. **77**, 3865 (1997)] in Vienna Ab Initio Simulation Package (VASP) [J. Phys. Rev. B **54**, 11169 (1996)]. The projector augmented-wave (PAW) pseudopotential [Phys. Rev. B **50**, 17953 (1994); Phys. Rev. B **71**, 391 (2005)] was employed for treating the atomic core electrons, and the plane-wave cutoff energy was set to 450 eV. In addition, the Van der Waals (vdW) correction was included with the Grimme DFT-D3 method [J. Chem. Phys. **132**, 154104 (2010); J. Comp. Chem. **32**, 1456 (2011)] for deriving the more accurate CO adsorption energy, more details can be seen in the Supplementary Information. Simulated STM topographic images of domain taken in an energy range of -1~1 eV. When considering the electric field, we optimized the adsorption structures and derived the energies by setting E-field = 0.00 eV/Å, 0.10 eV/Å, 0.20 eV/Å, and 0.35 eV/Å, respectively.

In the simulation models of adsorption structures, four layers of Au atoms were employed to mimic the Au(111) surface. The substrate was simulated by a hexagonal supercell with ($\sqrt{13}\times\sqrt{13}$)R±13.9° in-plane supercells (containing 7 adsorbed CO molecules). A 2×2×1 Monkhorst-Pack k-mesh was used to sample the first 2D Brillouin zone. During the geometry optimization, the two uppermost layers and the CO molecules were relaxed and the lower two layers of Au atoms were fixed, and a vacuum region of ≥ 15.0 Å was used in order to exclude periodic surface–surface interactions. The CO molecular films were initially constructed according to the STM images, where the CO molecules are placed vertically and ~ 3.0 Å above the Au(111) surface, and intermolecular distance between CO molecules was in the range of 3.7 to 4.0 Å. All the adsorption systems were fully relaxed until reaching force criterion of < 0.02 eV/Å.

Based on vibrational frequency analysis [J. Phys. Chem. C 2010, **114**, 18182–18197], the zero-point energy (ZPE) correction and entropy (ΔS) were also calculated to derive the adsorption free energy (ΔG) through

$$\Delta ZPE = ZPE_{nCO|Au111} - ZPE_{Au111} - n \cdot ZPE_{CO(g)} \quad (5)$$

$$\Delta S = S_{nCO|Au111} - S_{Au111} - n \cdot S_{CO(g)} \quad (6)$$

$$\Delta E_{ads} = E_{nCO|Au111} - E_{Au111} - n \cdot E_{CO(g)} \quad (7)$$

$$\Delta G = (\Delta E_{ads} + \Delta ZPE - T\Delta S)/n \quad (8)$$

where ΔE_{ads} is the adsorption energy of CO molecules on Au(111) surface, and ΔZPE and ΔS are respectively the differences of the zero-point energy and the entropy of the system before and after adsorption, n is the number of CO molecules on Au(111) surface.”

In addition, the adsorption energies under different electric fields were also considered in the revised manuscript (Page 7 Line 137-147):

“We further considered the adsorption free energies under the electric fields induced by the STM tip. Referring to the experimental electric field value (1.0 ~ 3.5 V/nm), we optimized the adsorption structures and derived the energies by setting E-field at 0.10 eV/Å, 0.20 eV/Å, and 0.35 eV/Å, respectively. As listed in Table 1, adsorption free energy ΔG of the cluster phase is gradually reduced with the increase of the applied electric field, by contrast, ΔG of the uniform phase demonstrates the increasing trend with the field strength. Such variation trend implies that the relative stabilities of the cluster phase and the uniform phase are respectively lowered and increased by the high applied electric field. Therefore, the DFT calculations

have confirmed that the transition between the cluster and the uniform phases could be induced the action of high electric field from the STM tip.”

Table 1. Adsorption free energy ΔG (T = 5 K) of different adsorption structures derived from DFT (unit: eV).

Structure model		E-field (eV/Å)			
		0.00	0.10	0.20	0.35
Cluster phase	Left-handed	-0.432	-0.430	-0.428	-0.426
	Right-handed	-0.432	-0.430	-0.429	-0.426
Uniform phase	Left -rotated	-0.379	-0.383	-0.386	-0.390
	Right -rotated	-0.379	-0.389	-0.392	-0.396

C3: 3) discuss that the value 0.007 eV ~ 0.7 kJ/mol is less than the usual numerical error;

R4: Thanks for the reviewer’s comments. We agree that the tiny energy difference is usually smaller than the systematic error of DFT method, thus, the absolute value of such tiny adsorption energy may be not meaningful. However, in our opinion, the relative energy order derived from DFT can still qualitatively show the relative stability of the phase structures.

Although we have included the dispersion correction in the DFT calculation, the difference between adsorption energy from DFT and the real adsorption energy could be still larger than ~ 0.7 kJ/mol. On the other hand, the error of DFT is usually systematic, where the DFT calculation can always overestimate or underestimate the adsorption energy of CO in similar systems. Here, we agree that the absolute values of the adsorption energies of -0.007 eV may be not very accurate, however, the energy difference between the cluster phase and uniform phase can be reasonable and may qualitatively reflect the relative stability between them. In experimental observation, the uniform phase is a highly mobile homogeneous phase, indicating its lower stability with small adsorption energy. The calculated adsorption energy of the uniform phase (~ -0.007 eV) is quite smaller than that of the cluster phase (-0.238 eV). The relative order of the calculated adsorption energies is consistent to the experimental observation.

Furthermore, in the revised manuscript, we further optimized the structure and obtained the energy using the DFT-D3 method. And we have replaced the adsorption energies to the adsorption free energies with the formula $\Delta G = \Delta E_{\text{ads}} + \Delta ZPE - T\Delta S$, which include the contributions from zero-point energy and the entropy, as list in the revised Table 1. It is found that because the experiments were carried out under a low temperature (~5 K), the temperature-induced effect is small, but the zero-point energy correction is important. According to the data in the third column of Table 1, the adsorption energy of the cluster phases is -0.43 eV, while that of the uniform phase is -0.39 eV. The conclusion that the cluster phase is more stable than the uniform phase still stands, and the adsorption energy of the uniform phases was corrected to -0.39 eV, which is significantly larger than -0.007 eV.

C4: 4) estimate the effect of the electric field on the energy barriers;

R4: Thanks for the referee's nice suggestion. We agree that the electric field should be considered in the diffusion barrier.

We have added the electric field into the DFT calculations, as shown in Supplementary Fig. 10. We chose the diffusion modes shown in Supplementary Fig. 10a to characterize the effect of the electric field on the barrier under the electric fields of $E = 0.10 \text{ eV/\AA}$ (Supplementary Fig. 10c) and 0.20 eV/\AA (Supplementary Fig. 10d), respectively. It is found that the applied electric field can apparently lower the diffusion barrier of CO. As shown in Supplementary Fig. 10d, with the electric field increased to 0.20 eV/\AA , the minimum barrier is decreased from 0.085 to 0.058 eV , indicating that the electric field induced by the STM tips can promote the motion of CO molecules. As a result, the lowered diffusion barrier under the applied electric fields demonstrates that the STM tip at high bias can help to induce the molecular diffusion as well as the phase transition.

According to the above analysis, we added this description in the revised manuscript (Page 8 Line 157-161):

“It is noteworthy that the diffusion barrier can be apparently reduced by the electric field (see Supplementary Fig. 10). With the electric field increased from 0.00 to 0.20 eV/\AA , the minimum barrier is decreased from 0.085 to 0.058 eV , indicating that the electric field induced by the STM tips can promote the motion of CO molecules, which can make the phase transition easier to occur.”

Supplementary Fig. 10: Calculated diffusion barriers of CO molecules on Au(111) surface. a, Structural models for estimating the diffusion barriers of CO molecules. Red, blue, and grey balls represent oxygen, carbon, and gold atoms, respectively. The DFT-calculated diffusion barriers without electric field (b), with the electric field of 0.10 eV/\AA (c), and with the electric field of 0.20 eV/\AA (d), respectively.

C5: 5) describe the non-linear amplification processes, for example, the authors cited the Frank model, but

there are no discussions and kinetic equations;

R5: We thank the referee for the suggestion. The chiral amplification phenomenon discussed in our manuscript is more likely a “soldier and sergeants” process driven by interface energy that can be described by the modified Frank model.

“Frank model is a naturally stochastic method, which assume that the initial production of small enantiomeric excess is a rare event, or initial chiral symmetry breaking is favored if statistical fluctuation is present. Then the environment is modified to make whole system into nonequilibrium states, which induces the generation of other chirality and results in the nonlinear chiral amplification [Biochim. Biophys. Acta. 1953, **11**: 459; Chem. Phys. Lett. **431**, (2006), 185; Phys. Chem. Chem. Phys. 2017, **19**, 29424; Chirality 2022, **34**, 104]. The model can be expressed by a simple system consisting of two chiral autocatalytic reaction between achiral and chiral (L and D enantiomers) reagents (with k_1 rate constant) coupled with a terminal competition reaction between them producing an achiral inhibition product with k_2 rate constant):

$$\frac{dn_L}{dt} = (k_1 - k_2 n_D) n_L \quad (1)$$

$$\frac{dn_D}{dt} = (k_1 - k_2 n_L) n_D \quad (2)$$

where n_L (n_D) is the concentrations of L (D) enantiomer. Through numeric calculation, it is given that

$$ee(t) \equiv n_L - n_D = (n_{L0} - n_{D0}) e^{k_1 t} \quad (3)$$

$$P_D \equiv P(ee) = (1 + e^{\alpha ee})^{-1} \quad (4)$$

where α depends on the specific system. As a result, the excess of one chirality over the other (ee) increases exponentially, while the probability of the minority to form homochiral states P_D decreases exponentially with the initial enantiomeric excess mediated by statistic fluctuation, as shown in Supplementary Fig. 12.”

Supplementary Fig. 12: The process of chiral amplification under stochastic Frank model. Solid curves are $P(ee)$, inset shows the evolution of ee over time. Parameters were chosen with $k_1 = k_2 = 0.5$, $\alpha = 21.48$, $N_D = 1.1$ (Ref from Phys. Chem. Chem. Phys. 2017, **19**, 29424).

The above statements have been added into the revised manuscript (Page 12).

C6: *6) More details about the experiment, mainly concerning the process of turning the field on and off, time, and periodicity. The work apparently indicates that only one cycle was applied concerning the electric field, i.e., field off, field on, and field off. However, for the amplification of the enantiomeric excess, it would take more cycles.*

R6: Thanks for the referee's suggestion. In our experiments, the high electric field is applied through scanning the area of CO adlayer with high bias voltage. The typical scanning time for one image is 5-7 min, corresponding to about 7-8 ms per pixel, which is long enough to complete the phase transition. In one operation process or one cycle, we scanned an area with CO cluster phases at low bias voltage. After one image is obtained, we set the bias to a higher value, corresponding to the high field on, and scanned the same area to induce the cluster phases transition to uniform phase. When the image at higher bias is obtained, the bias is set to the lower value, corresponding to the high field off, and we rescanned the same area to check the result after phase transition.

We agree the referee's opinion that more cycles are needed to achieve the amplification of the ee with the larger area. According the referee's suggestion, we performed the operation process containing multiple cycles on a larger area ($50 \times 50 \text{ nm}^2$) of CO cluster phases, as shown in Fig. 5, in which each STM image represent the change of cluster phases after one operation cycle at high voltage. We can notice that the size of domain B shrinks gradually, and finally the whole area turn into one domain A with homochirality, suggesting the chiral amplification occurs.

Fig. 5 The evolution of the homochirality of CO cluster phase on Au(111). Each image is scanned at low bias (-1.0 V, 6 pA) after one-time scanning at high bias (-3.5 V, 6 pA). Gray arrows represent the operation sequence. Yellow dotted lines marked the domain boundaries. Scale bars are 10 nm.

In revised manuscript, we added the details of experiments including the operation process for applying high electric field, and the example of operation process containing multiple cycles (Methods and Fig. 5).

C7: Another item that was not clear was the ramp of the electric field activation.

R7: As mentioned in **R6**, the electric field is applied through scanning the area of CO adlayer with high bias voltage. After one image is obtained at lower bias, we set the bias to a higher value with a ramp of ~300 mV/ms.

C8: Some minor issues are: (1) The title must indicate the electric field;

R8: Thank the referee's advices. In fact, the formation and amplification of homochirality is not directly related to the electric field. The role of electric field is to destroy the stable racemic structure (cluster phase) and make whole system into nonequilibrium/metastable state (uniform phase), thus inducing the generation of small enantiomeric excess during recondensation process and resulting in chiral amplification. Therefore, according to the requirement of length limitation by Nature Communication, we didn't indicate the electric field in the title, but we emphasized the role of electric field in Abstract.

C9: 2) In author contributions: Who is H.P. Li? Heping Li?

R9: Sorry for the mistake. The abbreviation of “Heping Li” should be “H. Li”. But this abbreviation will confuse with the author Hui Li. Therefore, we didn’t use the abbreviation of “Heping Li” in revised manuscript.

C10: 3) *it is necessary to explain the notation of chiral symmetry phases ($\sqrt{13}\times\sqrt{13}$)R \pm 13.9° adding primary references;*

R10: Thank the referee for the advice. We explained the notation of chiral symmetry phases should be ($\sqrt{13}\times\sqrt{13}$)R \pm 13.9° with respect to Au(111) 1 \times 1 lattice in revised manuscript (Line 98).

C11: 4) *To show a better crystallographic description for fcc and hcp;*

R11: Thanks for the referee’s suggestion. Due to the existence of strain, Au(111) surface exhibit $\sqrt{3}\times 22$ reconstruction with respect to 1 \times 1 lattice, which was also named as herringbone structure. According to different stacking sequence of top layer of Au atoms with respect to the underlayer, the alternating wide and narrow flat regions of herringbone structure are called as fcc and hcp parts, respectively. Such description is well accepted in the previous researches on the Au(111) surface. In order to introduce the adsorption sites of CO on Au(111) to the readers better, we also the adopt such description in our manuscript (Line 72-73).

C12: 5) *There are any formatting mistakes in the manuscript.*

R12: Sorry for the formatting mistakes. We have done our best to check and correct these mistakes in the revised manuscript.

C13: *I am sending a brief discussion about the kinetic equations based on the Frank model to help to improve the work. Of course, this is only a guide to help, and I am not sure about the correctness of these models in describing the experiment. There is a pleiad of possibilities of models and theories. The results are from differential equations, but it is better to consider stochastic methods to include the fluctuations. The left figure below is the usual Frank model; the central figure is a modified version of the Frank model. The final homochiral state is bounded following the experiment discussed here. As an observation, I do not know if this particular modified version is discussed in the literature. Finally, in the right figure, the modified Frank model was divided into two parts, one for the absence of the electric field and the other for the presence of the field. There are many possible descriptions, e.g., in the absence of the field, it is possible to use all the reactions of the modified Frank model. However, in the presence of the field, one can consider only the last step or other termination reactions. I have tested those, and the results are similar. It is possible to modify several parameters to describe the experiment better. (contains attached figure)*

R14: We extremely thank the referee for the valuable information, which help us to understand the Frank model better. Based on the stochastic description of modified Frank model, it's possible to give the chiral amplification of a small ee, even the formation of the homochirality of the minority. As far as we know, the Frank model under electric field have not been reported. The referee provided the calculated results of modified Frank model with the presence of electric field. However, in our experiment, the formation and amplification of homochirality is not directly related to the electric field. The role of electric field is to destroy the stable racemic structures and make the system into nonequilibrium states. The subsequent generation of small ee and chiral amplification in recondensation process happen after the high electric field withdraws. Therefore, we think it should belong to the common modified Frank model in the absence of the electric field.

Reply to referee #3:

C: *The paper reports the observation of chiral superstructures in the saturated adsorption layer of CO molecules on Au(111). The electric field from an STM tip can switch the chirality of CO heptamer clusters and induce chirality transition of the racemic molecular domains. The behavior is explained by DFT calculations that compare the thermodynamic stabilities of the corresponding structures. It is intriguing to find the chiral amplification phenomena in a simple system like CO/Au(111), which is a commonly studied system, nevertheless the evidence seems to be largely ignored. An in-depth understanding of the process may have general implications for the adsorption induced-chirality on surfaces.*

I would like to ask the authors to clarify some issues that are not clear in the article.

R1: Thank the referee for comments and suggestions on our work. In the following, we will address all the questions point-by-point.

C1: *1. More details of the DFT calculations should be provided. For example, the adsorption energies of hcp regions versus fcc regions and their electric field-dependency are absent. For a polar adsorbate, the effect of electric field polarity needs to be carefully discussed.*

R1: Thanks very much for the referee's comment. We agree that we did not give sufficient description of the DFT calculation details in the previous manuscript. Here, we have added more details to the calculation method in revised manuscript.

First, according to the reviewer's suggestions, we added more details of the DFT calculations in the revised manuscript. (Page 15, Theoretical calculations)

"The DFT calculations were carried out with Perdew, Burke, and Ernzerhof (PBE) functional [Phys. Rev. Lett. **77**, 3865 (1997)] in Vienna Ab Initio Simulation Package (VASP) [J. Phys. Rev. B **54**, 11169 (1996)]. The projector augmented-wave (PAW) pseudopotential [Phys. Rev. B **50**, 17953 (1994); Phys. Rev. B **71**, 391 (2005)] was employed for treating the atomic core electrons, and the plane-wave cutoff energy was set to 450 eV. In addition, the Van der Waals correction was included with the Grimme DFT-D3 method [J. Chem. Phys. **132**, 154104 (2010); J. Comp. Chem. **32**, 1456 (2011)] for deriving the more accurate CO adsorption energy, more details can be seen in the Supplementary Information. Simulated STM topographic images of domain taken in an energy range of -1~1 eV. When considering the electric field, we optimized the adsorption structures and derived the energies by setting E-field = 0.00 eV/Å, 0.10 eV/Å, 0.20 eV/Å, and 0.35 eV/Å, respectively.

In the simulation models of adsorption structures, four layers of Au atoms were employed to mimic the Au(111) surface. The substrate was simulated by a hexagonal supercell with $(\sqrt{13}\times\sqrt{13})R\pm 13.9^\circ$ in-plane supercells (containing 7 adsorbed CO molecules). A $2\times 2\times 1$ Monkhorst-Pack k-mesh was used to sample the first 2D Brillouin zone. During the geometry optimization, the two uppermost layers and the CO molecules were relaxed and the lower two layers of Au atoms were fixed, and a vacuum region of ≥ 15.0 Å was used in order to exclude periodic surface-surface interactions. The CO molecular films were initially constructed according to the STM images, where the CO molecules are placed vertically and ~ 3.0 Å above the Au(111) surface, and intermolecular distance between CO molecules was in the range of 3.7 to 4.0 Å. All the adsorption systems were fully relaxed until reaching force criterion of < 0.02 eV/Å.

Based on vibrational frequency analysis [J. Phys. Chem. C 2010, **114**, 18182–18197], the zero-point energy (ZPE) correction and entropy (ΔS) were also calculated to derive the adsorption free energy (ΔG)

through

$$\Delta ZPE = ZPE_{nCO|Au111} - ZPE_{Au111} - n \cdot ZPE_{CO(g)} \quad (5)$$

(6)

$$\Delta S = S_{nCO|Au111} - S_{Au111} - n \cdot S_{CO(g)}$$

(7)

$$\Delta E_{ads} = E_{nCO|Au111} - E_{Au111} - n \cdot E_{CO(g)}$$

$$\Delta G = (\Delta E_{ads} + \Delta ZPE - T\Delta S)/n \quad (8)$$

where ΔE_{ads} is the adsorption energy of CO molecules on Au(111) surface, and ΔZPE and ΔS are respectively the differences of the zero-point energy and the entropy of the system before and after adsorption, n is the number of CO molecules on Au(111) surface.”

Second, in terms of adsorption energies under the action of electric fields, we added this description in the revised manuscript (Page 7 Line 137-147):

“We further considered the adsorption free energies under the electric fields induced by the STM tip. Referring to the experimental electric field value (1.0 ~ 3.5 V/nm), we optimized the adsorption structures and derived the energies by setting E-field at 0.10 eV/Å, 0.20 eV/Å, and 0.35 eV/Å, respectively. As listed in Table 1, adsorption free energy ΔG of the cluster phase is gradually reduced with the increase of the applied electric field, by contrast, ΔG of the uniform phase demonstrates the increasing trend with the field strength. Such variation trend implies that the relative stabilities of the cluster phase and the uniform phase are respectively lowered and increased by the high applied electric field. Therefore, the DFT calculations have confirmed that the transition between the cluster and the uniform phases could be induced the action of high electric field from the STM tip.”

Table 1. Adsorption free energy ΔG (T = 5 K) of different adsorption structures derived from DFT (unit: eV).

Structure model		E-field (eV/Å)			
		0.00	0.10	0.20	0.35
Cluster phase	Left-handed	-0.432	-0.430	-0.428	-0.426
	Right-handed	-0.432	-0.430	-0.429	-0.426
Uniform phase	Left -rotated	-0.379	-0.383	-0.386	-0.390
	Right -rotated	-0.379	-0.389	-0.392	-0.396

Moreover, we have compared the adsorption energies of CO on the fcc and hcp sites of Au(111) surface under the electric field condition, as listed in Table R1. It can be seen that the electric field can slightly change the adsorption energies, but the more stable site for adsorption is still the fcc site.

Table R1. Energies of one CO adsorbed on Au (111) calculated by DFT (unit: eV).

E-field	0 eV/Å	0.1 eV/Å	0.2 eV/Å
---------	--------	----------	----------

Hcp-site	-0.532	-0.540	-0.547
Fcc-site	-0.553	-0.562	-0.570

C2: 2. The FFT spots in Fig.1c and Fig.3d-f are hardly visible, and it appears that Fig.1c is distorted. How can it be so sure that the poor contrast in Fig.3c is not due to the loss of STM resolution at high bias?

R2: Thanks for the referee's comments and sorry for the inconvenience. We have uploaded the electronic-version figures with high resolution separately to the submission system for the better review. The distortion in the original Fig. 1c is due to thermal drift during the scanning process, which can't be completely avoided in STM experiments. But the key features in Fig. 1c are still clear, and the data analysis is not strongly influenced.

In STM image of Fig. 3c taken at high bias, we can still observe the obvious spots corresponding to CO monomers (Enlarged image is shown in Fig. R2), which size is much smaller than CO heptamers observed in Fig. 3a taken at low bias. Therefore, the STM resolution is not loss at high bias. The lower contrast of STM image at high bias stem from the metastable uniform phase. The CO monomers in this phase will be easy to diffuse on the surface, causing the blurred feature on some area of surface. What's more, another group of FFT images under different bias is shown in Supplementary Fig. 2. Not only the CO monomer but the refined structure of herringbone is clearly observed, which can rule out the loss of resolution of the tip under high bias.

Fig. R2 Enlarged image (a) and corresponding FFT image (b) of Figure 3c. Green and blue arrows (circles) mark the orientation of CO monomer resolution of left- and right-rotated domains Scale bars are 2 nm in (a) and 2 nm^{-1} in (b).

Supplementary Fig. 2: The evolution of bias-dependent FFT images obtained from the STM images taken on the same area. Scanning parameter: $V_{tip} = -0.1$ V for (a); 2.5 V for (b); 3.0 V for (c) with the same current 5 pA. Scale bars are 2 nm⁻¹.

C3: 3. *I'm confused about "Although the left-handed and right-handed cluster phases can be formed on the whole Au(111) surface at low bias voltage". Are the chiral domains spontaneously formed or subjected to STM imaging?*

R3: Sorry for confusion. The left-handed and right-handed chiral domains are spontaneously formed on Au(111) surface. We revised the statement in line 174 as "the left-handed and right-handed cluster phases spontaneously formed and coexist on the surface due to stable adsorption energies".

C4: 4. *It is oversimplified to calculate the diffusion barrier with the model of a CO trimer on a bare substrate. The molecules in the assembly have a coherent movement to switch their chirality.*

R4: Thanks for the referee's comments. Firstly, we have added the diffusion barrier with the model of a CO heptamer, as shown in Supplementary Fig. 8d. The diffusion barrier of CO heptamer is apparently larger than it of the CO trimer, due to the more intermolecular interaction pairs around an individual CO molecule. On the other hand, such increase of the diffusion barrier should decay to zero for very large CO cluster, since the intermolecular Van de Waals interaction is a short-range interaction.

Supplementary Fig. 8: The potential energy surface of CO diffusion on Au(111). The structural models (a-d) and corresponding diffusion barriers (e-h) for CO molecules on Au (111) surface. Red, blue, and grey balls represent oxygen, carbon, and gold atoms, respectively. Red, blue, and green arrows indicate the diffusion directions of the CO molecules, corresponding to the red, blue, and green curves in (e-h).

Subsequently, we evaluate the barrier for the phase transition process using a 2×2 supercell of the heptamer structure, as shown in Supplementary Fig. 9. According to the structures of the left- and right-handed phases (Supplementary Fig. 9a and 9b), we can assume the first step of the phase transition path, in which the heptamer marked by the yellow circle is fixed, and the heptamer marked by the red circle passes two Cu atoms of the substrate to achieve the phase transition. As shown in Supplementary Fig. 9c, the process of the cluster moving from site-a to site-c experiences the transition from the left-handed phase to the right-handed phase. We scanned the potential energy surface along such phase transition path, and the result is shown in Supplementary Fig. 9d. It can be seen that point-b in the potential energy surface corresponds to the center of the transition process. The entire diffusion barrier is ~ 1.39 eV, corresponding to the diffusion barrier of ~ 0.20 eV for each CO molecule. Such value is consistent to the diffusion barrier shown in

Supplementary Fig. 8h.

Supplementary Fig. 9: Simulated phase transition model. **a, b,** Atomic models of the left- and right-handed cluster phases. **c,** Diagram simulating the phase transition process of a left-handed cluster to a right-handed cluster. The pathway of the red cluster moving from point-a to point-c on Au(111). **d,** Corresponding potential energy surface (PES) along the scanning path in (c). The calculated potential barrier is about 1.39 eV.

According to the above analysis, we added this description in the revised manuscript (Page 7 Line 148-157):

“Furthermore, the phase transition can be achieved through the diffusion of CO molecules on the substrate. The estimated minimum barrier of diffusion of single CO molecule on Au(111) is ~ 0.055 eV (see Supplementary Fig. 8a). Such diffusion barrier can be remarkably increased with the increase of cluster size. The diffusion barrier of CO in trimer and heptamer clusters are ~ 0.085 and ~ 0.152 eV (Supplementary Fig. 8b and 8d), respectively. In addition, the entire barrier for the transition from left-handed phase to the right-handed phase with a heptamer model is estimated to be ~ 1.39 eV (Supplementary Fig. 9d), consistent to the diffusion barrier of each CO molecule shown in Supplementary Fig. 8h.”

C5: *What is the threshold voltage for a chirality transition of domain? Is it possible to change the chirality of one or a few adjacent heptamer clusters in a racemic domain?*

R5: Thanks for referee’s question.

The threshold voltages for the transition of chiral cluster domain to a uniform phase varies as tunneling current is summarized in Supplementary Fig. 11. Red and black data point are from the threshold voltages under positive and negative tip bias respectively. And black solid line is fitted by linear function by formula $U \propto E \cdot (-\ln I)$. For the case of positive tip bias, the threshold voltage decreases as the tunneling current increases, indicating the transition from cluster phase to uniform phase is stimulated by electric field between

tip and sample. It is because the direction of electric field is parallel to the dipole moment of CO molecule on Au(111). The fitted value of critical field is about 0.79 V/\AA . Whereas, for the case of negative tip bias, corresponding to the direction of electric field is antiparallel to molecular dipole moment, the threshold voltage almost keeps same as tunneling current varies. We suppose the phase transition at negative bias is stimulated by electronic excitation or vibration excitation.

Supplementary Fig. 11: The threshold voltage required in the phase transition from cluster phase to uniform phase. a, Histogram showing the dependence of E_{th} on tunneling current. **b,** By making the logarithm of the tunnelling current, linear relationship is fitted by formula $U \propto E \cdot (-\ln I)$. The discrete points come from the experimental statistics and the solid lines are the corresponding linear fitting results. Error bars are about 0.15 V for all experimental data.

In our experiments, the high electric field is applied through scanning the area of CO adlayer with high bias voltage. In one operation process or one cycle, we scanned an area with CO cluster phases at low bias voltage firstly. After one image is obtained, we set the bias to a higher value, corresponding to the high field on, and scanned the same area to induce the cluster phases transition to uniform phase. When the image at higher bias is obtained, the bias is set to the lower value, corresponding to the high field off, and we rescanned the same area to check the result after phase transition. Such operation process is performed on the whole scanning area. In fact, we also tried to change the chirality of a few clusters near the boundaries of a racemic domain by applying voltage pulse. Supplementary Figure 7 shows a group of STM images which are obtained by successive scanning a same area at low bias voltage after several pulses of high bias voltage are applied. We can notice the chirality of clusters at the position of pulsing is changed, resulting in the domain boundary shorter. After several operation cycles of pulsing, almost the whole area covered by one domain with homochirality, suggesting the chirality amplification.

Supplementary Fig. 7: The evolution of domain boundaries after operation processes by bias pulsing. Each image is scanned at the low bias (-50 mV, 5 pA). The red circles mark the tip position of a pulse with 2.50 ± 0.25 V and 1s duration. Gray arrows represent the operation sequence. Yellow dotted lines marked the domain boundaries. Scale bars are 6 nm.

It should be emphasis that the formation and amplification of homochirality is not directly related to the electric field. Just as mentioned before, asymmetric synthesis is not possible if system have reached complete thermodynamic equilibrium [Chem. Rev. **98**, 2391 (1998)]. The role of electric field is to destroy the stable racemic structure (cluster phase) and make the system into nonequilibrium/metastable state (uniform phase), thus inducing the generation of small enantiomeric excess during recondensation process, resulting in chiral amplification.

C6: 5. *I would like to see a comparison of the original and after-electric field images of the same area to validate the proposed mechanism of domain boundary-driven amplification.*

R6: Thanks for referee's question.

As we mentioned before, in our experiments, the high electric field is applied through scanning the area of CO adlayer with high bias voltage. In one operation process or one cycle, we scanned an area with CO cluster phases at low bias voltage. After one image is obtained, we set the bias to a higher value, corresponding to the high field on, and scanned the same area to induce the cluster phases transition to uniform phase. When the image at higher bias is obtained, the bias is set to the lower value, corresponding to the high field off, and we rescanned the same area to check the result after phase transition.

In order to better illustrate the boundary-driven amplification, we show a group of STM images recording successive phases transition on a larger area ($50 \times 50 \text{ nm}^2$) of CO cluster phase with two or three cluster domains (see Supplementary Fig. 4), in which each image is scanned at low bias voltage after one-time scanning at high voltage. It is obvious that the boundaries between domain A and B gradually become shorter

and straighter, and the size of domain A became larger, reflecting the chiral amplification of CO cluster phase. Another example of operation process containing multiple cycles on a larger area ($50 \times 50 \text{ nm}^2$) of CO cluster phases is shown in Fig. 5, in which each STM image on the same area reveals the change of cluster phases after one operation cycle at high voltage. We can notice that the length of boundaries is shortened and the size of domain B shrinks gradually. Finally, the whole area turns into one domain A with homochirality, suggesting the chiral amplification occurs.

The driving force of the chiral amplification is the free energy of domain boundaries. In order to minimize the free energy of whole system, the domain boundaries prefer to become shorter and straighter, which induce the growth of domain with net prochirality. Moreover, Roman Fasel *et al* also pointed out that boundaries between mirror domains are energetically unfavorable than that between domains with prochirality [Nature **439**, 449 (2006)]. The “boundary-driven chiral amplification” based on statistical fluctuation is consistent with the modified Frank model, and has been reported in [Nature **439**, 449 (2006); Chem. Phys. Lett. **431** (2006) 185], which simulated the chiral separation under different local random fluctuations induced by reaction-noise.

Supplementary Fig. 4: The successive evolution of domain boundaries between two kinds of domains of CO cluster phase on Au(111). Each image is scanned at low bias (-0.1 V, 5 pA) after one-time scanning at high bias (-3.5 V, 5pA). Gray arrows represent the operation sequence. Yellow dotted lines marked the domain boundaries. Scale bars are 8 nm.

Fig. 5 The evolution of the homochirality of CO cluster phase on Au(111). Each image is scanned at low bias (-1.0 V, 6 pA) after one-time scanning at high bias (-3.5 V, 6 pA). Gray arrows represent the operation sequence. Yellow dotted lines marked the domain boundaries. Scale bars are 10 nm.

C7: *The hcp and fcc regions should be clearly indicated. If only fcc regions can change their chirality under electric field, will the hcp regions remain intact after this perturbation?*

R7: Thanks for the referee's comment. Due to the existence of strain, Au(111) surface exhibit $\sqrt{3}\times 22$ reconstruction with respect to 1×1 lattice, which was also named as herringbone structure. According to different stacking sequence of top layer of Au atoms with respect to the underlayer, the alternating wide and narrow flat parts of herringbone structure are called as fcc and hcp regions, respectively. We have indicated the exact position of fcc and hcp regions on Figure 1 and 4 in revised manuscript.

The fcc regions of herringbone structure have higher surface reactivity than hcp regions due to larger atomic spacing and the center of d states [Phys. Rev. Lett. **81** (1998) 2819-2822], resulting in higher adsorption stability of foreign atoms or molecules on fcc region than hcp region [J. Chem. Phys. **149**, 034703 (2018); Top Catal. **54**, 1357–1367 (2011); Phys. Rev. B **89**, 195425 (2014)]. In our experiments, the role of electric field is to destroy the stable racemic structure (cluster phase) and make the system into metastable states (uniform phase), thus inducing the generation of enantiomeric excess and resulting in chiral amplification. Since the CO clusters on fcc regions are more stable than those on hcp regions, when the uniform phase recondenses to cluster phase, the cluster phase starts to nucleate on the fcc region, and diffuses over the surface. Therefore, the chirality of cluster phase on the hcp regions is dependent on the process of

chiral amplification, and may not remain intact after the operation of high electric field.

C8: 6. *I cannot understand the argument of "asymmetry amplification is both kinetically feasible and thermodynamically favorable". What is the kinetic factor in this case?*

R8: Thanks for the referee's question.

Firstly, it is easy to understand the asymmetry amplification is thermodynamically favorable. The mono-chirality structure has much smaller domain boundary area, which has higher energy than the cluster phase. Asymmetry amplification can stabilize the adsorption system through eliminating the high-energy domain boundary area. Thus, asymmetry amplification is thermodynamically favorable.

On the other hand, the asymmetry amplification process is also kinetically feasible. The asymmetry amplification process contains two steps: the cluster phase is firstly induced to the uniform phase, and then the system is annealed back to the cluster phase. Such structure transition can be achieved by CO diffusion on the substrate. Our DFT calculation shows that the higher electric field can significantly lower the diffusion barrier of CO (see Supplementary Fig. 10), so that the diffusion coefficient of CO is drastically increased with the help of high electric field. Therefore, the phase transition is kinetically feasible under high electric field.

According to the above analysis, we added this description in the revised manuscript (Page 7 Line 137-170):

"We further considered the adsorption free energies under the electric fields induced by the STM tip. Referring to the experimental electric field value (1.0 ~ 3.5 V/nm), we optimized the adsorption structures and derived the energies by setting E-field at 0.10 eV/Å, 0.20 eV/Å, and 0.35 eV/Å, respectively. As listed in Table 1, adsorption free energy ΔG of the cluster phase is gradually reduced with the increase of the applied electric field, by contrast, ΔG of the uniform phase demonstrates the increasing trend with the field strength. Such variation trend implies that the relative stabilities of the cluster phase and the uniform phase are respectively lowered and increased by the high applied electric field. Therefore, the DFT calculations have confirmed that the transition between the cluster and the uniform phases could be induced the action of high electric field from the STM tip.

...It is noteworthy that the diffusion barrier can be apparently reduced by the electric field (see Supplementary Fig. 10). With the electric field increased from 0.00 to 0.20 eV/Å, the minimum barrier is decreased from 0.085 to 0.058 eV, indicating that the electric field induced by the STM tips can promote the motion of CO molecules, which can make the phase transition easier to occur.

To sum up, the DFT calculations show that the cluster phase has lower free energy than the uniform phase, so that the domain boundary area, which can be considered to have similar uniform adsorption structure, is less stable than the cluster structure. To stabilize the adsorption system through eliminating the high-energy domain boundary area, the asymmetry amplification process is thermodynamically favorable. Furthermore, the present DFT calculations also confirm that the high electric field induced from the STM tip can significantly lower the diffusion barrier and drastically increase the diffusion coefficient of CO molecules on the substrate. As a result, the structure transition is also more kinetically feasible under high electric field."

Table 1. Adsorption free energy ΔG (T = 5 K) of different adsorption structures derived from DFT (unit: eV).

Structure model	E-field (eV/Å)
-----------------	----------------

		0.00	0.10	0.20	0.35
Cluster phase	Left-handed	-0.432	-0.430	-0.428	-0.426
	Right-handed	-0.432	-0.430	-0.429	-0.426
Uniform phase	Left -rotated	-0.379	-0.383	-0.386	-0.390
	Right -rotated	-0.379	-0.389	-0.392	-0.396

Supplementary Fig. 10: Calculated diffusion barriers of CO molecules on Au(111) surface. **a**, Structural models for estimating the diffusion barriers of CO molecules. Red, blue, and grey balls represent oxygen, carbon, and gold atoms, respectively. The DFT-calculated diffusion barriers without electric field (**b**), with the electric field of 0.1 eV/\AA (**c**), and with the electric field of 0.2 eV/\AA (**d**), respectively.

Reply to referee #4:

C: Liu *et al.* report in the manuscript „Condensation and asymmetric amplification of chirality in achiral molecules on an achiral surface” results obtained using scanning tunneling microscopy (at liquid He temperature) and density functional theory on the formation of chiral (mirror-symmetric) phases formed upon clustering or domain forming CO adsorbed on Au(111) – the latter surface apparently in its known Herringbone reconstruction.

The authors show by Fourier transforming STM micrographic images that long-range ordered areas or domains of CO clusters (6 or 7mers) are formed. The electric field imposed by the STM tip (using bias voltage of ca. 1 V) induces CO diffusion and the two observed domains (A and B or left- and right-handed) “dissolve” and re-assemble again, with in fact complete suppression of one of the phases (left-handed only, as described within lines 125 – 141). The authors also refer to the Extended Data Fig. 1. The authors conclude on the so-called chiral amplification in this phase transition, meaning that one of the “chiral phases” dominates – a so-called enantiomeric excess occurs. DFT calculations using the Perdew, Burke, Ernzerhof generalized-gradient approximation to electronic exchange and correlation effects were employed to calculate adsorption energies (on DFT-level, i.e. total electronic energy differences only) of the two phases in comparison to a so-called uniform CO adsorption phase on the ideal, non-reconstructed Au(111) surface model.

R: Thank the referee’s comments. We will address all the questions in the following point-by-point.

C1: 1) How do the authors explain that only the left-handed or left-rotated form of the CO clusters are formed in their chiral amplification experiments. How many times have these experiments carried out and are these experiments reproducible?

R1: Thank the referee’s comments.

In our system, the stable cluster phases of CO adlayer with chirality can transition to metastable uniform phase by high electric field. After withdrawing the high electric field, the metastable uniform phase is recondensed into cluster phases. During this recondensation process, the statistic noise fluctuation will break the chiral symmetry and induce the spatial segregated domains separated by racemic boundaries, which result in the formation of enantiomeric excess. The domain with homochirality (left-handed or right-handed) will be grown further by successive operation process at high electric field, revealing the chiral amplification. The driving force of the chiral amplification is the free energy of domain boundaries. In order to minimize the free energy of whole system, the domain boundaries prefer to become shorter and straighter, which induce the growth of domain with homochirality. Moreover, Roman Fasel *et al* also pointed out that boundaries between mirror domains are energetically unfavorable than that between domains with prochirality [Nature **439**, 449 (2006)]. The “boundary-driven chiral amplification” based on statistical fluctuation is consistent with the modified Frank model, and has been reported in [Nature **439**, 449 (2006); Chem. Phys. Lett. **431** (2006) 185], which simulated the chiral separation under different local random fluctuations induced by reaction-noise.

In our work, we carried out the experiments hundreds of times, and the results are reproducible. After the multiple process cycles of phase transition, only left-handed or right-handed cluster phase remains on the whole scanning area. Here, we are showing two extra examples to indicate the chiral amplification in our system. Figure 5 contains a group of STM images containing multiple operation cycles of phase transition on a larger area (50×50 nm²) of CO cluster phases. Each STM image on the same area reveals the change of

cluster phases after one operation cycle at high voltage. We can notice that the length of domain boundaries is shorten and the size of domain B shrinks gradually. Finally, the whole area turns into one domain A with left-handed chirality, suggesting the chiral amplification occurs. Supplementary Figure 5 also contains a group of STM images recording successive operation of phase transition on a larger area ($40\times 40\text{ nm}^2$) of CO cluster phases. It is obvious that the boundaries between domain A and B gradually become shorter and straighter, and the size of domain B with right-handed chirality became larger and cover the whole area finally, reflecting the chiral amplification.

We speculate that the chiral amplification here is driven by the domain boundary free energy under statistic fluctuation. Therefore, for the initial racemic condition, the formation probability of homochirality of left- and right-handed cluster phase should be equal. For example, Figure 5 and Supplementary Figure 5 show the generation of almost homochirality for left-hand and right-hand, respectively.

Fig. 5 The evolution of the homochirality of CO cluster phase with domain A on Au(111). Each image is scanned at low bias (-1.0 V, 6 pA) after one-time scanning at high bias (-3.5 V, 6 pA). Gray arrows represent the operation sequence. Yellow dotted lines marked the domain boundaries. Scale bars are 10 nm.

Supplementary Fig. 5: The evolution of the homochirality of CO cluster phase with domain B on Au(111). Each image is scanned at low bias (-1.0 V, 5 pA) after scanning at a positive high bias. Gray arrows represent the operation sequence. Yellow dotted lines marked the domain boundaries. Scale bars are 8 nm.

C2: 2) *The Herringbone reconstruction of Au substrate is clearly visible on the STM images in Fig. 1. The CO adsorption – in my opinion- follows the $(22 \times \sqrt{3})$ reconstruction. To what extent – as the “thermodynamic window” or energy difference-window is narrow (see Table 1 and main text) – do the authors think that this reconstruction does NOT affect their observation. The reason I ask is the work by Hanke and Björk (PRB 87, 235422 (2013)), who showed that “neglect” of the reconstruction – especially the different straight and elbow areas involving different coordination numbers of Au atoms – affects adsorption energies.*

R2: Thanks for the referee’s comments.

We agree that the herringbone reconstruction of Au(111) has a significant effect on the adsorption of CO molecules. It’s well known that the fcc region in Au(111) surface tends to have higher surface reactivity due to larger atomic spacing and the center of d states [Phys. Rev. Lett. **81** (1998) 2819-2822], resulting in the higher adsorption stability of atoms or molecules than hcp region [J. Chem. Phys. **149**, 034703 (2018); Top Catal (2011) **54**:1357–1367; Phys. Rev. B **89**, 195425 (2014)]. However, in our experiments, the Au(111) surface is fully covered by CO adlayer. The domains of CO cluster phases with both kind of chirality can extend dozens of nanometers, and cross several fcc and hcp regions as well as even elbow areas. Therefore, we believe the influence from different regions of herringbone on the formation of cluster phases is negligible.

In our experiment, the electric field is applied to destroy the stable racemic structure (cluster phase) and make the system into metastable states (uniform phase), thus inducing the generation of enantiomeric excess and resulting in chiral amplification. Due to the small difference in adsorption energy between fcc and hcp regions, as the electric field is set to a moderate value (1.5 V), the cluster phases on hcp regions transition to uniform phase while those on fcc regions keep unchanged (Fig. 4b). If the bias voltage increases to a very high value (3.5 V), the cluster phases on all regions are destroyed. When we withdraw the high electric field, the uniform phase recondenses to cluster phase. At that time, the cluster phase starts to nucleate on the fcc region firstly (Fig. 4d), and diffuses over the surface. The domains of cluster phase with homochirality can

extend to the whole scanning area (Figure 5). Therefore, though the fcc regions may act as nucleation sites during recondensation process, we think the reconstruction of Au(111) should not affect the formation of cluster phases with homochirality, as well as our observation of chiral amplification.

Moreover, we have carefully supplemented the adsorption of CO on fcc and hcp sites of Au(111) surface under the electric field condition, as shown in the Table R1. It is obvious that the most stable site for adsorption of CO is fcc site. However, as we discussed above, such difference in adsorption energy can not influence the formation of cluster phases as well as the chiral amplification.

Table R1. Energies of one CO adsorbed on Au (111) calculated by DFT (unit: eV).

E-field	0 eV/Å	0.1 eV/Å	0.2 eV/Å
Hcp-site	-0.532	-0.540	-0.547
Fcc-site	-0.553	-0.562	-0.570

C3: 3) I must criticize that the authors did not care about van der Waals type of dispersion effects.

R3: Thanks for the referee's comments. We have carefully checked the vdW corrections in the DFT calculation in the revised manuscript.

According to the reviewer's suggestion, we added the following instruction in the revised manuscript: (Page 15 Line 327-329 Methods: Theoretical calculations)

“In addition, the Van der Waals correction was included with the Grimme DFT-D3 method [J. Chem. Phys. **132**, 154104 (2010); J. Comp. Chem. **32**, 1456 (2011)] for deriving the more accurate CO adsorption energy, more details can be seen in the Supplementary Information.”

And, we added the following instruction in the revised Supporting Information (Page 9):

“We have compared the adsorption energies without and with various dispersion correction functions (PBE [Phys. Rev. Lett. **77**, 3865 (1997)], DFT-D2 [J. Comp. Chem. **27**, 1787 (2006)], DFT-D3 [J. Chem. Phys. **132**, 154104 (2010); J. Comp. Chem. **32**, 1456 (2011)], and optB86b [J. Phys.: Cond. Matt. **22**, 022201 (2010)]), as listed in Supplementary Table 1. It can be seen that PBE without any vdW correction significantly underestimate the adsorption energy of CO. Therefore, the vdW-correction is necessary for the CO adsorption calculation. On the other hand, DFT-D2, DFT-D3, and optB86b give very consistent adsorption energy values (-0.4 ~ -0.5 eV), indicating that the mainstream dispersion correction methods are all suitable for the present systems. Here, we employed the most widely used DFT-D3 method as the dispersion correction function in the present work.”

Supplementary Table 1. The calculated adsorption energy considering different Van der Waals types (unit: eV).

vdWs type	Structure model	$E_{\text{ads}}/7\text{CO}$	E_{ads}/CO
PBE	Cluster phase	Left-handed	-1.054
		Right-handed	-1.060

	Uniform phase	Left-rotated	-0.812	-0.116
		Right-rotated	-0.797	-0.114
optB86b	Cluster phase	Left-handed	-2.971	-0.424
		Right-handed	-2.973	-0.425
	Uniform phase	Left-rotated	-2.726	-0.389
		Right-rotated	-2.711	-0.387
DFT-D2	Cluster phase	Left-handed	-3.632	-0.519
		Right-handed	-3.634	-0.519
	Uniform phase	Left-rotated	-3.537	-0.505
		Right-rotated	-3.562	-0.511
DFT-D3	Cluster phase	Left-handed	-3.150	-0.450
		Right-handed	-3.153	-0.450
	Uniform phase	Left-rotated	-2.838	-0.405
		Right-rotated	-2.826	-0.404

C4: 4) As the experiments are carried out close to the absolute zero of temperature – T -induced effects are supposed to be small, but one should even then care about zero-point energy corrections. The DFT-modelling part is in this respect a bit too “coarse grained” given the very small energy differences the authors are aiming for to reproducibly describe and discuss.

R4: We thank the referee very much for the comment. We agree that the DFT-modelling part is in this respect a bit too “coarse grained”. We should explain the calculation method in detail and take into account the effect of temperature. Therefore, we have revised Table 1 and added the corresponding discussion in the revised manuscript, which can more accurately demonstrate the adsorption stability of CO on Au(111).

First, we added the following instruction of the computational details in the revised manuscript. (Page 15, Methods: Theoretical calculations)

“The DFT calculations were carried out with Perdew, Burke, and Ernzerhof (PBE) functional [Phys. Rev. Lett. **77**, 3865 (1997)] in Vienna Ab Initio Simulation Package (VASP) [J. Phys. Rev. B **54**, 11169 (1996)]. The projector augmented-wave (PAW) pseudopotential [Phys. Rev. B **50**, 17953 (1994); Phys. Rev. B **71**, 391 (2005)] was employed for treating the atomic core electrons, and the plane-wave cutoff energy was set to 450 eV. In addition, the Van der Waals correction was included with the Grimme DFT-D3 method [J. Chem. Phys. **132**, 154104 (2010); J. Comp. Chem. **32**, 1456 (2011)] for deriving the more accurate CO adsorption energy, more details can be seen in the Supplementary Information. Simulated STM topographic images of domain taken in an energy range of $-1 \sim 1$ eV. When considering the electric field, we optimized the adsorption structures and derived the energies by setting E-field = 0.00 eV/Å, 0.10 eV/Å, 0.20 eV/Å, and 0.35 eV/Å, respectively.

In the simulation models of adsorption structures, four layers of Au atoms were employed to mimic the Au(111) surface. The substrate was simulated by a hexagonal supercell with $(\sqrt{13}\times\sqrt{13})R\pm 13.9^\circ$ in-plane supercells (containing 7 adsorbed CO molecules). A $2\times 2\times 1$ Monkhorst-Pack k-mesh was used to sample the first 2D Brillouin zone. During the geometry optimization, the two uppermost layers and the CO molecules were relaxed and the lower two layers of Au atoms were fixed, and a vacuum region of $\geq 15.0 \text{ \AA}$ was used in order to exclude periodic surface-surface interactions. The CO molecular films were initially constructed according to the STM images, where the CO molecules are placed vertically and $\sim 3.0 \text{ \AA}$ above the Au(111) surface, and intermolecular distance between CO molecules was in the range of 3.7 to 4.0 \AA . All the adsorption systems were fully relaxed until reaching force criterion of $< 0.02 \text{ eV/\AA}$.

Based on vibrational frequency analysis [J. Phys. Chem. C 2010, **114**, 18182–18197], the zero-point energy (ZPE) correction and entropy (ΔS) were also calculated to derive the adsorption free energy (ΔG) through

$$\Delta ZPE = ZPE_{n\text{CO|Au111}} - ZPE_{\text{Au111}} - n \cdot ZPE_{\text{CO(g)}} \quad (5)$$

$$\Delta S = S_{n\text{CO|Au111}} - S_{\text{Au111}} - n \cdot S_{\text{CO(g)}} \quad (6)$$

$$\Delta E_{\text{ads}} = E_{n\text{CO|Au111}} - E_{\text{Au111}} - n \cdot E_{\text{CO(g)}} \quad (7)$$

$$\Delta G = (\Delta E_{\text{ads}} + \Delta ZPE - T\Delta S)/n \quad (8)$$

where ΔE_{ads} is the adsorption energy of CO molecules on Au(111) surface, and ΔZPE and ΔS are respectively the differences of the zero-point energy and the entropy of the system before and after adsorption, n is the number of CO molecules on Au(111) surface.”

Second, according to the reviewer’s suggestions, we have revised Table 1 and added the corresponding discussion in the revised manuscript (Page 7 Line 135-147):

“The adsorption free energies (ΔG) of both the cluster and the uniform phase structures derived from the DFT calculations are listed in Table 1. The adsorption free energies of the cluster and uniform phases are about -0.43 eV and -0.39 eV, respectively. We further considered the adsorption free energies under the electric fields induced by the STM tip. Referring to the experimental electric field value (1.0 ~ 3.5 V/nm), we optimized the adsorption structures and derived the energies by setting E-field at 0.10 eV/ \AA , 0.20 eV/ \AA , and 0.35 eV/ \AA , respectively. As listed in Table 1, adsorption free energy ΔG of the cluster phase is gradually reduced with the increase of the applied electric field, by contrast, ΔG of the uniform phase demonstrates the increasing trend with the field strength. Such variation trend implies that the relative stabilities of the cluster phase and the uniform phase are respectively lowered and increased by the high applied electric field. Therefore, the DFT calculations have confirmed that the transition between the cluster and the uniform phases could be induced the action of high electric field from the STM tip.”

Table 1. Adsorption free energy ΔG (T = 5 K) of different adsorption structures derived from by DFT (unit: eV).

Structure model		E-field (eV/ \AA)			
		0.00	0.10	0.20	0.35
Cluster phase	Left-handed	-0.432	-0.430	-0.428	-0.426
	Right-handed	-0.432	-0.430	-0.429	-0.426

Uniform phase	Left -rotated	-0.379	-0.383	-0.386	-0.390
	Right -rotated	-0.379	-0.389	-0.392	-0.396

C5: 5) *The transition states for CO diffusion have been optimized using which methods? These aspects were not described in the manuscript. Is the diffusion energy barrier of 0.07 eV numerically converged? Is there really one imaginary mode along the diffusion path – I mean the barrier is close to the numerical error bars or uncertainties which are technically attainable.*

R5: Thanks very much for the referee’s advice. We are sorry that we did not describe our method of estimating the barrier in the manuscript. Usually, the NEB method is a stricter method for the transition state research, but here, we employed a more direct way to scan the diffusion potential energy surface (PES) to derive the diffusion barrier with reference to previous literature [Int. Lett. Chem. 2014, **38**: 26-34; Spect. Acta A 2013, **115**: 64-73; Comput. Mater. Sci. 2018, **146**:287-302]. The relaxed potential energy surface scanning is a common method and widely used in study reaction mechanisms, locating transition structures, studying bond stability, and evaluating conformational flexibility, especially for the surface diffusion problem [Inorg. Chem. 2004, **43**: 1116-1121; J. Chem. Theory & Comput. 2012, **8**: 1870; J. Phys. Chem. B 2012, **116**: 5860-5871; J. Am. Chem. Soc. 2006, **128**:1287-1292].

We believe that the 0.07 eV we have gotten can be considered convergent because our scanning accuracy is 0.2 angstroms per scan point. For the structure corresponding to each scanning point, we fixed the x and y direction of C, relaxed the z direction of C and the xyz direction of O atom, and each structure optimization reached force criterion of < 0.02 eV/Å. Then, according to the calculated data points, we fit the curve of the potential energy surface. Finally, we estimate a barrier of 0.07 eV. Through calculation simulation, we try our best to guess the CO possible diffusion path, and estimate the barrier on each path, although we are estimated to get the value.

“Is there really one imaginary mode along the diffusion path?” Since to calculate the frequencies and normal mode analysis for such complicated surface system is too difficult, we did not check the imaginary mode here. We must admit that the scanning PES method can only roughly estimate the diffusion barrier, but we believe that the diffusion path and transition barrier here should be not far from the exact transition state.

According to the reviewer’s suggestions, we added the following instruction of methods for CO diffusion barrier in the revised Supporting Information. (Page 9, Methods of calculate the diffusion barrier)

“The diffusion barrier was estimated through scanning the potential energy surface (PES) of CO diffusing on the substrate. The relaxed PES scanning is a common method and widely used in study reaction mechanisms, studying bond stability, evaluating conformational flexibility, and locating transition structures especially for the surface diffusion problem. [Int. Lett. Chem. 2014, **38**: 26-34; Spect. Acta A 2013, **115**: 64-73; Comput. Mater. Sci. 2018, **146**:287-302; Inorg. Chem. 2004, **43**: 1116-1121; J. Chem. Theory & Comput. 2012, **8**: 1870; J. Phys. Chem. B 2012, **116**: 5860-5871; J. Am. Chem. Soc. 2006, **128**:1287-1292]. For the structure at each scanning point, the x and y direction of carbon atoms was fixed, while the z direction of carbon atoms and the xyz directions of oxygen atoms were fully relaxed. The force criterion of structure optimization was 0.02 eV/Å.”

C6: *Overall, I cannot recommend this manuscript for publication in Nature Communications.*

R6: As shown in the replies above, we added more experimental data to exhibit our experiment is reproducible, and performed DFT calculations by considering vdWs effect and zero-point energy corrections. The influences from Au(111) herringbone structure and calculation details on diffusion barriers were also discussed. We believe our results in experiments and theoretical calculations are reliable, and hope the referee can give a more positive judgement.

Reviewers' Comments:

Reviewer #2:

Remarks to the Author:

The work describes cluster phases consisting of chiral CO heptamers. This stable initial racemic cluster phase can be transformed into a metastable uniform phase of CO monomers by applying an electric field (interacting with the electric molecular dipole). After withdrawing the electric field, the cluster phase is recovered with an enantiomeric excess. A chiral amplification is obtained after successive electric field applications to reach a possible homochiral cluster phase.

The manuscript improved significantly after the first round of review. The authors responded satisfactorily to my questions. However, some issues still need to be clarified.

1) Usually, it is only necessary to show calculations with dispersion since it is presumable to use dispersion correction between molecular systems. I presume that DFT-D2 and DFT-D3 are dispersion corrections using the PBE functional. It is important to say explicitly in the text;

2) Where is the barrier (Fig.8h) as a function of the electric field for the heptamer? You showed the trimer result to clarify the importance of the electric field in the barrier. How difficult is it to get this result for the heptamer?

3) The ordinate in Fig.9d in SI is not correct;

4) Organize the figures in sequence;

5) Avoid text repetitions on the manuscript and also between the manuscript and SI;

6) The original model by Frank [48] is deterministic. The essential differences between Eqs.(3) [48] and (4) [49] are: Eq.(3) is analytic and describes the evolution of $ee(t)$ strictly towards the non-zero initial $ee(t=0)$. On the other hand, Eq.(4) is numerically obtained using a stochastic method, and the probability is given in any direction from the initial $ee(t=0)$, i.e., any homochiral state can be obtained respecting the probability given by Eq.(4). These must be fixed in the text.

Reviewer #3:

Remarks to the Author:

The authors acceptably address most of my questions and concerns. However, there are two points that I would like to stress again.

1.The authors suggest that the phase transition at negative bias could be stimulated by electronic excitation or vibration excitation. However, this issue is ignored in the manuscript, and misleadingly claims that both negative and positive bias voltage could induce phase transition in a CO/Au system.

2.To answer my question 4, the authors calculated the diffusion barrier of a CO heptamer, which is more relevant to the experimental observation in the study. However, this diffusion barrier of CO heptamer clusters is estimated to be ~ 0.152 eV, which seems to be significant for a moderate tip voltage to induce a phase transition.

Reviewer #4:

Remarks to the Author:

After careful reading the rebuttal, the substantially extended and amended supplementary material and the amended manuscript I am inclined to recommend publication of the revised manuscript. The

authors indeed clarified the comments and criticisms of all the reviewers. The topic and work is interesting and the quality of the work and the manuscript warrants publication (although the theory part still operates at the edge of numerical error bars).

Reply to referee #2:

C: *The work describes cluster phases consisting of chiral CO heptamers. This stable initial racemic cluster phase can be transformed into a metastable uniform phase of CO monomers by applying an electric field (interacting with the electric molecular dipole). After withdrawing the electric field, the cluster phase is recovered with an enantiomeric excess. A chiral amplification is obtained after successive electric field applications to reach a possible homochiral cluster phase.*

The manuscript improved significantly after the first round of review. The authors responded satisfactorily to my questions. However, some issues still need to be clarified.

R: Thanks for the referee's suggestions and support. We carefully address all the questions in the following point-by-point.

C1: *1) Usually, it is only necessary to show calculations with dispersion since it is presumable to use dispersion correction between molecular systems. I presume that DFT-D2 and DFT-D3 are dispersion corrections using the PBE functional. It is important to say explicitly in the text;*

R1: Thanks for the referee's suggestions. We would like to apologize for incorrectly referring to the dispersion as DFT-D3 in the "Method" of our previous manuscript. As the referee pointed out, our calculations were based on the PBE functional, and therefore, we have now made the necessary correction by replacing "DFT-D3" with "PBE-D3" in both the manuscript and Supplementary Information.

C2: *2) Where is the barrier (Fig.8h) as a function of the electric field for the heptamer? You showed the trimer result to clarify the importance of the electric field in the barrier. How difficult is it to get this result for the heptamer?*

R2: Thanks for the referee's suggestion. We have revised the manuscript and added Supplementary Figures 6 and 7. These figures demonstrate the diffusion barriers of CO under different electric field strengths for monomer, trimer and heptamer clusters. As shown in Supplementary Figure 6, the minimum barrier of CO in heptamer decreased from 0.152 eV to 0.138 eV as the electric field increased from 0.00 to 0.20 V/Å, which supports our conclusion that electric fields can reduce the diffusion barrier of CO. Additionally, Supplementary Fig. 7 shows the diffusion barriers of CO monomer, trimer and heptamer all follow the same trend with increasing electric field strength, where the stronger electric field lead to lower diffusion barriers.

Supplementary Fig. 6: Calculated diffusion barriers of CO in a heptamer cluster on Au(111) surface. **a**, Structural models for estimating the diffusion barriers of CO molecules. Red, blue, and grey balls represent oxygen, carbon, and gold atoms, respectively. The calculated diffusion barriers **(b)** without electric field, **(c)** with the electric field of 0.10 V/\AA , and **(d)** $E = 0.20 \text{ V/\AA}$, respectively. The ordinate are relative energies with the energy of the initial structure defined as “0”. The other data points on the curve represent the difference from the energy of the initial structure. The arrows marked the maximum diffusion barriers.

Supplementary Fig. 7: Diffusion barriers as a function of applied electric field. Black, red and blue show the diffusion barrier as a function of the electric field for CO in monomer, trimer and heptamer, respectively.

We added the following instruction into the revised Supporting Information (Page 7): “To further investigate the effect of electric field on CO diffusion, we selected the diffusion modes of CO in heptamer for the same calculation. Supplementary Figure 6 shows that, with the electric field increased from 0.00 to 0.20 V/Å, the minimum barrier of CO in heptamer decreased from 0.152 to 0.138 eV. This result confirms that the STM tip at high bias can induce molecular diffusion and facilitate the phase transition by lowering the diffusion barrier.”

C3: 3) *The ordinate in Fig.9d in SI is not correct;*

R3: Sorry for mistake in Supplementary Fig. 9d. After careful checking, we have corrected the ordinate as shown in revised Supplementary Fig. 4d. It should be noted that the ordinate represents relative energies, with the structural energy at point “a” as the reference energy. All the points later on the curve are relative to this reference energy. We have also improved the caption of Supplementary Fig. 4 with “The ordinate represents relative energies, with the structural energy at point “a” as the reference energy, and all the points on the curve are relative to this reference energy.”

Supplementary Fig. 4: Simulated phase transition model. a, b, Atomic models of the left- and right-handed cluster phases. **c,** Diagram simulating the phase transition process of a left-handed cluster to a right-handed cluster. The pathway of the red cluster moving from point-a to point-c on Au (111). **d,** Corresponding potential energy surface (PES) along the scanning path in (c). The calculated potential barrier is ~1.39 eV. The ordinate represents relative energies, with the structural energy at point “a” as the reference energy, and all the points on the curve are relative to this reference energy.

C4: 4) *Organize the figures in sequence; (exp.)*

R4: We appreciate the referee’s comment and agree that the Supplementary Information should be organized more conveniently to meet the requirements of the manuscript. Accordingly, we have adjusted the order of the figures as follows: “Fig. 1 S1 2 3 S2 Table 1 S3 S4 S5 S6 S7 4 S8 S9 S10 S11 5 S12 S13 S14 Table S1”. This should make it easier for readers to navigate the Supplementary Information.

C5: 5) Avoid text repetitions on the manuscript and also between the manuscript and SI; (exp.)

R5: We appreciate the referee's efforts to improve the clarity and conciseness of the manuscript and SI. In the revised manuscript and SI, the duplicate parts have been removed to improve readability.

C6: 6) The original model by Frank [48] is deterministic. The essential differences between Eqs.(3) [48] and (4) [49] are: Eq.(3) is analytic and describes the evolution of $ee(t)$ strictly towards the non-zero initial $ee(t=0)$. On the other hand, Eq.(4) is numerically obtained using a stochastic method, and the probability is given in any direction from the initial $ee(t=0)$, i.e., any homochiral state can be obtained respecting the probability given by Eq.(4). These must be fixed in the text.

R6: We are very sorry for lack of detail and overly ambitious model description in the original manuscript. We have added additional content to the "Discussion" part (Page 11) in the revised manuscript as following:

“The original Frank model constructed by F. C. Frank was deterministic system consisting of two chiral autocatalytic reactions between achiral and chiral (L and D enantiomers) reagents (with a rate constant of k_1), coupled with a terminal competition reaction between them that produce an achiral inhibition product (with a rate constant of k_2):

$$\frac{dn_L}{dt} = (k_1 - k_2 n_D) n_L \quad (1)$$

$$\frac{dn_D}{dt} = (k_1 - k_2 n_L) n_D \quad (2)$$

where n_L (n_D) is the concentrations of L (D) enantiomer. Through numeric calculation, it is given that:

$$ee(t) \equiv n_L - n_D = (n_{L0} - n_{D0}) e^{k_1 t} \quad (3)$$

The evolution of $ee(t)$ under original perturbation $ee(0)$ demonstrates that the final configuration of the system is strictly depend on its initial excess.

Systems in nature are subject to various disturbances at any given time. The Frank model can be modified to account for stochastic fluctuations, which cause the system to deviate slightly from its equilibrium states. Over time, two enantiomers may become equal in excess, or there may be a possibility for the enantiomer with an enantiomeric deficiency to become the dominant final state. We can define P_w as the probability of the enantiomer w (L or D) becoming the final homochiral state. There is

$$\frac{P_D}{P_L} = \lim_{t \rightarrow \infty} \frac{1}{M} \sum_{i=1}^M \frac{n_D(t)}{n_L(t)} = e^{-\alpha ee}. \quad (4)$$

Where M is the number of different stochastic trajectories. α is the coefficient depending on the specific system. Since $P_L + P_D = 1$,

$$P_D \equiv (1 + e^{\alpha ee})^{-1}. \quad (5)$$

As a result, the probability of forming homochiral states P_D decreases exponentially with any initial enantiomeric excess mediated by statistic fluctuation, highlighting the role of statistical fluctuations in the emergence of homochirality (see Supplementary Fig. 13).”

Reply to referee #3:

C: *The authors acceptably address most of my questions and concerns. However, there are two points that I would like to stress again.*

R: Thanks for the referee's comments. In the following, we will address all the questions point-by-point.

C1: *1. The authors suggest that the phase transition at negative bias could be stimulated by electronic excitation or vibration excitation. However, this issue is ignored in the manuscript, and misleadingly claims that both negative and positive bias voltage could induce phase transition in a CO/Au system.*

R1: We are very apologized for this misleadingly claim made in original manuscript. In the Supplementary Information, we have shown, for the case of negative tip bias, corresponding to the direction of electric field is antiparallel to molecular dipole moment, the threshold voltage for phase transition almost keeps same as tunneling current varies. Therefore, we suppose the phase transition at negative bias is stimulated by electronic excitation or vibration excitation. In the revised manuscript, we also have emphasized the phase transition in CO/Au system is induced by both high negative and positive bias voltages. The phase transition at the positive bias voltage should be stimulated by electric field, while the phase transition at the negative bias voltage should be stimulated by electric or vibration excitation.

C2: *2. To answer my question 4, the authors calculated the diffusion barrier of a CO heptamer, which is more relevant to the experimental observation in the study. However, this diffusion barrier of CO heptamer clusters is estimated to be ~0.152 eV, which seems to be significant for a moderate tip voltage to induce a phase transition.*

R2: Thanks for the referee's comments. Compared with the diffusion barrier of single CO and CO in trimer (~0.06 eV), the diffusion barrier of CO in heptamer is higher at 0.152 eV, indicating that the heptamer is much more stable than a monomer and trimer on substrate. However, the diffusion barrier of 0.152 eV is not sufficient to prevent CO diffusion under high electric field. It has been reported that CO molecules can easily cross the barrier of 0.32 eV on the Ni(211) surface [Appl. Surf. Sci. 399 (2017), 255–264.]. Additionally, the barrier can be further decreased from 0.152 eV to 0.138 eV with an increase of the electric field from 0.00 V/Å to 0.20 V/Å, as shown in Supplementary Fig. 6. Therefore, the calculated barrier is moderate for a tip voltage to induce easier diffusion and phase transition of CO molecules on Au(111).

In revised manuscript, we have modified the description about Supplementary Fig. 3 (Page 7): “we estimate the minimum diffusion barrier of CO monomer, CO in trimer and heptamer on Au(111), as shown in Supplementary Fig. 3. Compared with the diffusion barrier of single CO and CO in trimer (~0.06 eV), the diffusion barrier of CO in heptamer is higher at 0.152 eV, indicating that the heptamer is much more stable than a monomer and trimer on substrate. This barrier is less than that of CO molecules on other noble metal surface [Appl. Surf. Sci. 399 (2017), 255–264.], supporting an easier diffusion for a moderate tip voltage.”

And we added the following instruction into the revised Supplementary Information (Page 7-10): “To further investigate the effect of electric field on CO diffusion, we selected the diffusion modes of CO in heptamer for the same calculation. Supplementary Figure 6 shows that, with the electric field increased from 0.00 to 0.20 V/Å, the minimum barrier of CO in heptamer decreased from 0.152 to 0.138 eV. This result confirms that the STM tip at high bias can induce molecular diffusion and facilitate the phase transition by lowering the diffusion barrier.”

“Based on the above discussion, a quantitative dependence of the diffusion barriers on applied electric

field can be obtained. We calculated the diffusion barriers of CO monomer, CO in trimer and CO in heptamer with electric fields of 0.30 V/Å, 0.40 V/Å and 0.50 V/Å. Interestingly, we observed that the diffusion barriers of CO decrease linearly with the increase of electric field (Supplementary Fig. 7). This also confirms that the electric field can reduce the CO diffusion barrier and further promote the phase transition.”

Supplementary Fig. 6: Calculated diffusion barriers of CO molecules in heptamer on Au(111) surface. **a**, Structural models for estimating the diffusion barriers of CO molecules. Red, blue, and grey balls represent oxygen, carbon, and gold atoms, respectively. The DFT-calculated diffusion barriers without electric field (**b**), with the electric field of 0.10 V/Å (**c**), and with the electric field of 0.20 V/Å (**d**), respectively. The ordinate are relative energies, the energy of the initial structure is defined as “0”. The other data points on the curve represent the difference from the energy of the initial structure. The arrows marked the maximum diffusion barriers.

Supplementary Fig. 7: Diffusion barriers as a function of applied electric field. Black, red and blue show

the diffusion barrier as a function of the electric field for CO in monomer, trimer and heptamer, respectively.

Reply to referee #4:

C: *After careful reading the rebuttal, the substantially extended and amended supplementary material and the amended manuscript I am inclined to recommend publication of the revised manuscript. The authors indeed clarified the comments and criticisms of all the reviewers. The topic and work is interesting and the quality of the work and the manuscript warrants publication (although the theory part still operates at the edge of numerical error bars).*

R: We appreciate the referee's support of our work. While the referee may have concerns regarding the calculated values of diffusion barriers of single CO and CO in trimer (0.06 eV and 0.09 eV) at the edge of theoretical calculation error, we believe these values are reliable and effective. The diffusion barrier for atom or molecules on the metal surface is typically very small. For example, the diffusion barrier of atomic hydrogen on metals ranges from 0.02 eV to 0.40 eV [Appl. Surf. Sci. 2017, 420:1-8]. Similarly, the diffusion barriers for Au self-diffusion on Cl covered Au(001) surface range from 0.07 eV to 0.23 eV [J. Chem. Phys. 151, 064709 (2019)]. The diffusion energy of other systems under PBE path were also reported within the range of 0.05-0.20 eV [Phys. Rev. Lett. 110, 235901 (2013); Phys. Rev. B 90, 155208 (2014); Chem. Lett. 2014, 43,1940-1942; Comput. Mater. Sci. 140, (2017, 47-54)]. And meanwhile the barriers are usually underestimated with PBE function [Phys. Rev. Lett, 108, 256403 (2012)]. In this manuscript, without considering the effect of electric field, the calculated CO diffusion barrier falls within a reasonable range of 0.06 eV ~ 0.20 eV. Therefore, we believe that the numerical error of DFT for the similar systems should be systematic, and the numerical error does not affect the conclusions of the present work.

Reviewers' Comments:

Reviewer #2:

Remarks to the Author:

The manuscript improved significantly after the second round of review. The authors responded satisfactorily to my questions. Therefore, the manuscript is fine to be accepted.

Reviewer #3:

Remarks to the Author:

The authors fully answered my questions and made corresponding revision. I recommend the paper to be published in Nature Communications.

Reviewer #4:

Remarks to the Author:

I am inclined to recommend publication of the revised manuscript as it stands.

Reply to referee #2:

C: *The manuscript improved significantly after the second round of review. The authors responded satisfactorily to my questions. Therefore, the manuscript is fine to be accepted.*

R: We thank the reviewer for his/her support to our work.

Reply to referee #3:

C: *The authors fully answered my questions and made corresponding revision. I recommend the paper to be published in Nature Communications.*

R: We appreciate the referee very much.

Reply to referee #4:

C: *I am inclined to recommend publication of the revised manuscript as it stands.*

R: Thanks for the referee's recognition of our work and efforts.